# K-Radar: 4D Radar Object Detection for Autonomous Driving in Various Weather Conditions

**Dong-Hee Paek**[1][*]  **Seung-Hyun Kong**[1][*][†]  **Kevin Tirta Wijaya**[2]
[1]CCS Graduate School of Mobility  [2]Robotics Program
KAIST
{donghee.paek, skong, kevin.tirta}@kaist.ac.kr

## Abstract

Unlike RGB cameras that use visible light bands (384∼769 THz) and Lidars that use infrared bands (361∼331 THz), Radars use relatively longer wavelength radio bands (77∼81 GHz), resulting in robust measurements in adverse weathers. Unfortunately, existing Radar datasets only contain a relatively small number of samples compared to the existing camera and Lidar datasets. This may hinder the development of sophisticated data-driven deep learning techniques for Radar-based perception. Moreover, most of the existing Radar datasets only provide 3D Radar tensor (3DRT) data that contain power measurements along the Doppler, range, and azimuth dimensions. As there is no elevation information, it is challenging to estimate the 3D bounding box of an object from 3DRT. In this work, we introduce KAIST-Radar (K-Radar), a novel large-scale object detection dataset and benchmark that contains 35K frames of 4D Radar tensor (4DRT) data with power measurements along the Doppler, range, azimuth, and elevation dimensions, together with carefully annotated 3D bounding box labels of objects on the roads. K-Radar includes challenging driving conditions such as adverse weathers (fog, rain, and snow) on various road structures (urban, suburban roads, alleyways, and highways). In addition to the 4DRT, we provide auxiliary measurements from carefully calibrated high-resolution Lidars, surround stereo cameras, and RTK-GPS. We also provide 4DRT-based object detection baseline neural networks (baseline NNs) and show that the height information is crucial for 3D object detection. And by comparing the baseline NN with a similarly-structured Lidar-based neural network, we demonstrate that 4D Radar is a more robust sensor for adverse weather conditions. All codes are available at https://github.com/kaist-avelab/k-radar.

## 1 Introduction

An autonomous driving system generally consists of sequential modules of perception, planning, and control. As the planning and control modules rely on the output of the perception module, it is crucial for the perception module to be robust even under adverse driving conditions.

Recently, various works have proposed deep learning-based autonomous driving perception modules that demonstrate remarkable performances in lane detection (Paek et al., 2022; Liu et al., 2021), object detection (Wang et al., 2021a; Lang et al., 2019; Major et al., 2019), and other tasks (Ranftl et al., 2021; Teed and Deng, 2021). These works often use RGB images as the inputs to the neural networks due to the availability of numerous public large-scale datasets for camera-based perception. Moreover, an RGB image has a relatively simple data structure, where the data dimensionality is

---

[*]co-first authors

[†]corresponding author

36th Conference on Neural Information Processing Systems (NeurIPS 2022) Track on Datasets and Benchmarks.

relatively low and neighboring pixels often have high correlation. Such a simplicity enables deep neural networks to learn the underlying representations of images and recognize objects on the image.

Unfortunately, camera is prone to poor illumination, can easily be obscured by raindrops and snowflakes, and cannot preserve depth information that is crucial for accurate 3D scene understanding of the environment. On the other hand, Lidar actively emits measuring signals in the infrared spectrum, therefore, the measurements are hardly affected by illumination conditions. Lidar can also provide accurate depth measurements within centimeters resolution. However, Lidar measurements are still affected by adverse weathers since the wavelength of the signals ($\lambda$=850nm$\sim$1550nm) is not long enough to pass through raindrops or snowflakes (Kurup and Bos, 2021).

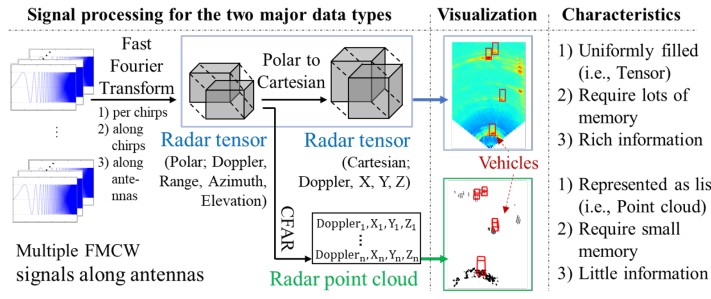

Figure 1: An overview of the signal processing of the FMCW Radar and a visualization of the two main data types (i.e., Radar tensor (RT) and Radar point cloud (RPC)). The RT is a dense data matrix with power measurements in all element along the dimensions through a Fast Fourier Transform (FFT) operation applied to FMCW signals. Since all elements are non-zero values, the RT provides dense information regarding the environment with minimal loss, at a cost of high memory requirement. On the other hand, the RPC is a data type in which target (i.e., object candidate group) information is extracted in the form of a point cloud with a small amount of memory by applying Constant False Alarm Rate (CFAR) algorithm to the RT. Due to the ease of implementing FFT and CFAR directly on the hardware, many Radar sensors provide RPCs as output. However, the RPC may lose a significant amount of information regarding the environment due to the CFAR algorithm.

Similar to Lidar, a Radar sensor actively emits waves and measures the reflection. However, Radar emits radio waves ($\lambda \approx 4$mm) that can pass through raindrops and snowflakes. As a result, Radar measurements are robust to both poor illumination and adverse weather conditions. This robustness is demonstrated in (Abdu et al., 2021), where a Frequency Modulated Continuous Wave (FMCW) Radar-based perception module is shown to be accurate even in adverse weather conditions and can be easily implemented directly on the hardware.

As FMCW Radars with dense Radar tensor (RT) outputs become readily available, numerous works (Dong et al., 2020; Mostajabi et al., 2020; Sheeny et al., 2021) propose RT-based object detection networks with comparable detection performance to camera and Lidar-based object detection networks. However, these works are limited to 2D bird-eye-view (BEV) object detection, since FMCW Radars utilized in existing works only provide 3D Radar tensor (3DRT) with power measurements along the Doppler, range, and azimuth dimensions.

In this work, we introduce KAIST-Radar (K-Radar), a novel 4D Radar tensor (4DRT)-based 3D object detection dataset and benchmark. Unlike the conventional 3DRT, 4DRT contains power measurements along the Doppler, range, azimuth, and elevation dimensions so that the 3D spatial information can be preserved, which could enable accurate 3D perception such as 3D object detection with Lidar. To the best of our knowledge, K-Radar is the first large-scale 4DRT-based dataset and benchmark, with 35k frames collected from various road structures (e.g. urban, suburban, highways), time (e.g. day, night), and weather conditions (e.g. clear, fog, rain, snow). In addition to the 4DRT, K-Radar also provides high-resolution Lidar point clouds (LPCs), surround RGB images from four stereo cameras, and RTK-GPS and IMU data of the ego-vehicle.

Since the 4DRT high-dimensional representation is unintuitive to human, we leverage the high-resolution LPC so that the annotators can accurately label the 3D bounding boxes of objects on the road in the visualized point clouds. The 3D bounding boxes can be easily transformed from the Lidar to the Radar coordinate frame since we provide both spatial and temporal calibration parameters to correct offsets due to the separations of the sensors and the asynchronous measurements, respectively. K-Radar also provides a unique tracking ID for each annotated object that is useful for tracking

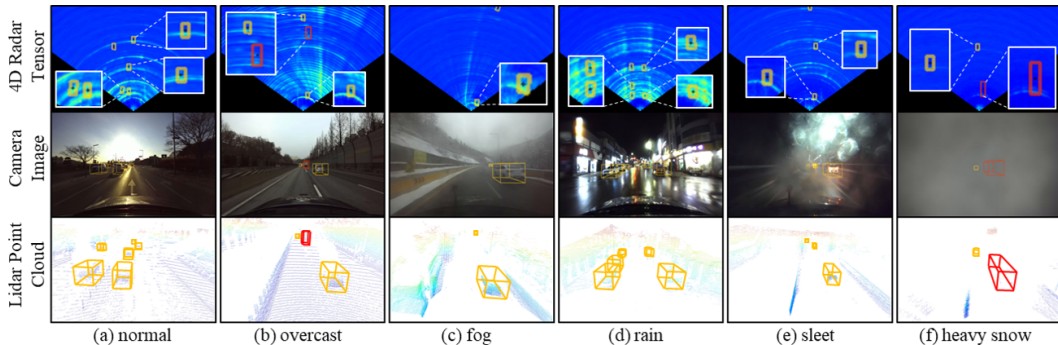

Figure 2: Samples of K-Radar datasets for various weather conditions. Each column shows (1) 4DRTs, (2) front view camera images, and (3) Lidar point clouds (LPCs) of different weather conditions. 4DRTs are represented in a two-dimensional (BEV) Cartesian coordinate system using a series of visualization processes that are described in Section 3.3. In this example, yellow and red bounding boxes represent the sedan and bus or truck classes, respectively. Appendix A contains further samples of K-Radar datasets for each weather condition.

an object along a sequence of frames. Examples of information regarding tracking are shown in Appendix I.7.

To demonstrate the necessity of 4DRT-based perception module, we present a 3D object detection baseline neural network (baseline NN) that directly consumes 4DRT as an input. From the experimental results on K-Radar, we observe that the 4DRT-based baseline NN outperforms the Lidar-based network in the 3D object detection task, especially in adverse weather conditions. We also show that the 4DRT-based baseline NN utilizing height information significantly outperforms network that only utilizes BEV information. Additionally, we publish the complete development kits (devkits) that include: (1) training / evaluation codes for 4DRT-based neural networks, (2) labeling / calibration tools, and (3) visualization tools to accelerate research in the field of 4DRT-based perception.

In a summary, our contributions are as follow,

- We present a novel 4DRT-based dataset and benchmark, K-Radar, for 3D object detection. To the best of our knowledge, K-Radar is the first large-scale 4DRT-based dataset and benchmark with diverse and challenging illumination, time, and weather conditions. With the carefully annotated 3D bounding box labels and multimodal sensors, K-Radar can also be used for other autonomous driving tasks such as object tracking and odometry.
- We propose a 3D object detection baseline NN that directly consumes 4DRT as an input and verify that the height information of 4DRT is essential for 3D object detection. We also demonstrate the robustness of 4DRT-based perception for autonomous driving, especially under adverse weather conditions.
- We provide devkits that include: (1) training/evaluation, (2) labeling/calibration, and (3) visualization tools to accelerate 4DRT-based perception for autonomous driving research.

The remaining of this paper is organized as follows. Section 2 introduces existing datasets and benchmarks that are related to perception for autonomous driving. Section 3 explains the K-Radar dataset and baseline NNs. Section 4 discusses the experimental results of the baseline NN on the K-Radar dataset. Section 5 concludes the paper with a summary and discussion on the limitations of this study.

## 2   Related Works

Deep neural networks generally require a large amount of training samples collected from diverse conditions so that they can achieve remarkable performance with excellent generalization. In autonomous driving, there are numerous object detection datasets that provide large-scale data of various sensor modalities, shown in Table 1.

Table 1: Comparison of object detection datasets and benchmarks for autonomous driving. HR and LR refer to High Resolution Lidar with more than 64 channels and Low Resolution with less than 32 channels, respectively. Bbox., Tr.ID, and Odom. refer to bounding box annotation, tracking ID, and odometry, respectively. Bold text indicates the best entry in each category.

| Data-set | Num. data | Sensors | | | | | Label | | |
|---|---|---|---|---|---|---|---|---|---|
| | | RT | RPC | LPC | Camera | GPS | Bbox. | Tr. ID | Odom. |
| K-Radar (ours) | 35K | **4D** | **4D** | **HR.** | **360.** | **RTK** | **3D** | **O** | **O** |
| VoD | 8.7K | X | **4D** | **HR.** | Front | **RTK** | **3D** | **O** | **O** |
| Astyx | 0.5K | X | **4D** | LR. | Front | X | **3D** | X | X |
| RADDet | 10K | 3D | 3D | X | Front | X | 2D | X | X |
| Zendar | 4.8K | 3D | 3D | LR. | Front | GPS | 2D | **O** | **O** |
| RADIATE | 44K | 3D | 3D | LR. | Front | GPS | 2D | **O** | **O** |
| CARRADA | 12.6K | 3D | 3D | X | Front | X | 2D | **O** | X |
| CRUW | 396K | 3D | 3D | X | Front | X | Point | **O** | X |
| NuScenes | 40K | X | 3D | LR. | **360.** | **RTK** | **3D** | **O** | **O** |
| Waymo | 230K | X | X | **HR.** | **360.** | X | **3D** | **O** | X |
| KITTI | 15K | X | X | **HR.** | Front | **RTK** | **3D** | **O** | **O** |
| BDD100k | **120M** | X | X | X | Front | **RTK** | 2D | **O** | **O** |

KITTI (Geiger et al., 2012) is one of the earliest and widely-used datasets for autonomous driving object detection that provide camera and Lidar measurements along with accurate calibration parameters and 3D bounding box labels. However, the number of samples and the diversity of the dataset is relatively limited since the 15K frames of the dataset are collected mostly in urban areas during daytime. Waymo (Sun et al., 2020) and NuScenes (Caesar et al., 2020) on the other hand provide a significantly larger number of samples with 230K and 40K frames, respectively. In both datasets, the frames are collected during both daytime and nighttime, increasing the diversity of the datasets. Additionally, NuScenes provides 3D Radar point clouds (RPC), and Nabati and Qi (2021) demonstrates that utilizing Radar as an auxiliary input to the neural network can improve the detection performance of the network. However, RPC lose a substantial amount of information due to the CFAR thresholding operation and result in inferior detection performance when being used as the primary input to the network. For example, the state-of-the-art performance of Lidar-based 3D object detection on NuScenes dataset is 69.7% mAP, whereas for Radar-based is only 4.9% mAP.

In the literature, there are several 3DRT-based object detection datasets for autonomous driving. CARRADA (Ouaknine et al., 2021) provides Radar tensors in the range-azimuth and range-Doppler dimensions with labels of up to two objects in a controlled environment (wide flat surface). Zenar (Mostajabi et al., 2020), RADIATE (Sheeny et al., 2021), and RADDet (Zhang et al., 2021) on the other hand provide Radar tensors collected on real road environments, but can only provide 2D BEV bounding box labels due to the lack of height information in 3DRTs. CRUW (Wang et al., 2021b) provides a large number of 3DRTs, but annotations only provide 2D point locations of objects. VoD (Palffy et al., 2022) and Asytx (Meyer and Kuschk, 2019) provide 3D bounding box labels with 4DRPCs. However, the dense 4DRTs are not made available, and the number of samples in the datasets is relatively small (i.e., 8.7K and 0.5K frames). To the best of our knowledge, the proposed K-Radar is the first large-scale dataset that provide 4DRT measurements on diverse conditions along with 3D bounding box labels.

Table 2: Comparison of object detection datasets and benchmarks for autonomous driving. d/n refers to day and night. Bold text indicates the best entry in each category.

| Dataset | Weather conditions | Time |
|---|---|---|
| K-Radar (ours) | **overcast, fog, rain, sleet, snow** | **d/n** |
| VoD | X | day |
| Astyx | X | day |
| RADDet | X | day |
| Zendar | X | day |
| RADIATE | overcast, fog, rain, snow | **d/n** |
| CARRADA | X | day |
| CRUW | X | day |
| NuScenes | overcast, rain | **d/n** |
| Waymo | overcast | **d/n** |
| KITTI | X | day |
| BDD100k | overcast, fog, rain, snow | **d/n** |

Autonomous cars should be capable to operate safely even under adverse weather conditions, therefore, the availability of adverse weather data in an autonomous driving dataset is crucial. In the literature, the BDD100K (Yu et al., 2020) and RADIATE datasets contain frames acquired under adverse weather conditions, as shown in Table 2. However, BDD100K only provides RGB front images, while RADIATE only provides 32-channel low-resolution LPC. Meanwhile, the proposed K-Radar provides 4DRT, 64-channel and 128-channel high-resolution LPC, and 360-degree RGB stereo images, which enables the development of multi-modal approaches using Radar, Lidar, and camera for various perception problems for autonomous driving under adverse weather conditions.

# 3 K-Radar

In this section, we describe the configuration of the sensors used to construct the K-Radar dataset, the data collection process, and the distribution of the data. Then, we explain the data structure of a 4DRT, along with the visualization, calibration, and labelling processes. Finally, we present 3D object detection baseline networks that can directly consume 4DRT as the input.

## 3.1 Sensor specification for K-Radar

To collect data under adverse weathers, we install five types of waterproofed sensors (listed in Appendix B) with IP66 rating, according to the configuration shown in Figure 3. First, a 4D Radar is attached to the front grill of the car to prevent multi-path phenomenon due to the bonnet or ceiling of the car. Second, a 64-channel Long Range Lidar and a 128-channel High Resolution Lidar are positioned at the centre of the car's roof with different heights (Figure 3-(a)). The Long-Range LPCs are used for accurately labelling objects of various distances, while the High-Resolution LPCs provide dense information with a wide (i.e., 44.5 degree) vertical field of view (FOV). Third, a stereo camera is placed on the front, rear, left, and right side of the vehicle, which results in four stereo RGB images that cover 360-degree FOV from the ego-vehicle perspective. Last, an RTK-GPS antenna and two IMU sensors are set on the rear side of the vehicle to enable accurate positioning of the ego-vehicle.

## 3.2 Data collection and distribution

The majority of frames with adverse weather conditions are collected in Gangwon-do of the Republic of Korea, a province that has the highest annual snowfall nationally. On the other hand, frames with

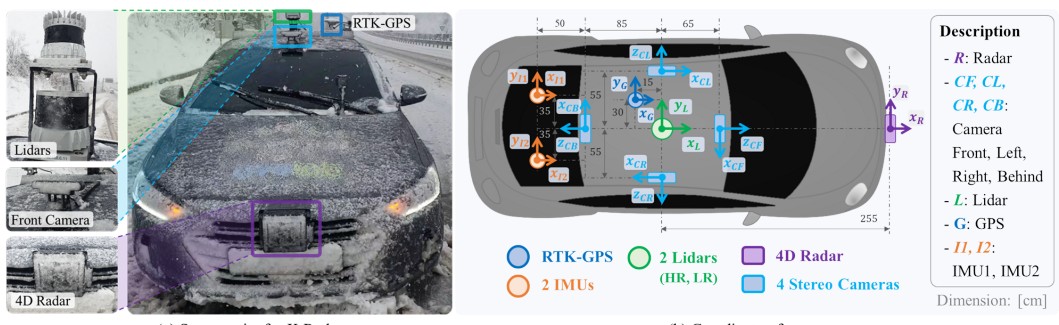

(a) Sensor suite for K-Radar (b) Coordinate of sensors

Figure 3: Sensor suite for K-Radar and coordinate system of each sensor. (a) shows the condition of the sensors after a 5 minute drive in heavy snow. Since the car drives forward, snow accumulates heavily in front of the sensors and covers the front camera lens, Lidar and Radar surfaces as shown in (a). As a result, during heavy snow, most of the information regarding the environment cannot be acquired by the front-facing camera and the Lidar. In contrast, Radar sensors are robust to adverse weathers, since the emitted waves can pass through raindrops and snowflakes. This figure emphasizes (1) the importance of Radar in adverse weather conditions, especially in heavy snowy conditions, and (2) the need for sensor placement and additional design (e.g., installation of wipers in front of the Lidar) considering the adverse weather conditions. (b) shows the installation location of each sensor and the coordinate system of each sensor.

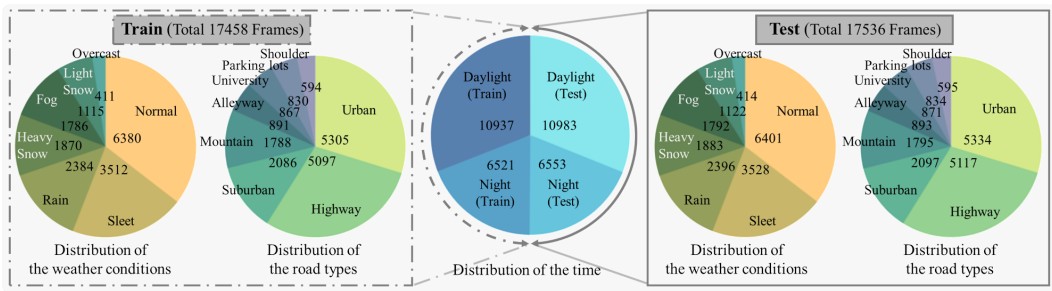

Figure 4: Distribution of data over collection time (night/day), weather conditions, and road types. The central pie chart shows the distribution of data over collection time, while the left and right pie charts show the distribution of data over weather conditions and road types for the train and test sets, respectively. At the outer edges of each pie chart, we state the collection time , weather conditions, and road types, and at the inner part, we state the number of frames in each distribution.

urban environments are mostly collected in Daejeon of the Republic of Korea. The data collection process results in 35K frames of multi-modal sensor measurements that constitute the K-Radar dataset. We classify the collected data into several categories according to the criteria listed on Appendix C. In addition, we split the dataset into training and test sets in a way that each condition appears in both sets in a balanced manner, as shown in Figure 4.

In total, there are 93.3K 3D bounding box labels for objects (i.e., sedan, bus or truck, pedestrian, bicycle, and motorcycle) on the road within the longitudinal radius of 120m and lateral radius of 80m from the ego-vehicle. Note that we only annotate objects that appear in the positive longitudinal axis, i.e., in front of the ego-vehicle.

In Figure 5, we show the distribution of object classes and object distances from the ego-vehicle in the K-Radar dataset. The majority of objects lie within 60m distance from the ego-vehicle, with 10K∼15K objects appearing in each of the 0m∼20m, 20m∼40m, and 40m∼60m distance categories, and around 7K objects appearing in over 60m distance category. As a result, K-Radar can be used to evaluate the performance of a 3D object detection networks for objects on various distances.

Figure 5: Objects classes and distance-to-ego-vehicle distribution for the training/test splits provided in the K-Radar dataset. We state the object class name and distance to the ego-vehicle in the outer part of the piechart, and the number of objects in each distribution in the inner part of the pie chart.

### 3.3 Data visualization, calibration, and annotation processes

Contrary to the 3D Radar tensor (3DRT) that lacks height information, 4D Radar tensor (4DRT) is a dense data tensor filled with power measurements in four dimensions: Doppler, range, azimuth, and elevation. However, the additional dimensionality of dense data imposes a challenge in visualizing 4DRT as a sparse data such as a point cloud (Figure 2). To cope with the problem, we visualize 4DRT as a two-dimensional heat map in the Cartesian coordinate system through heuristic processing as shown in Figure 6-(a), which results in 2D heatmap visualizations in the bird-eye-view (BEV-2D), front-view (FV-2D), and side-view (SV-2D). We refer to these 2D heatmaps collectively as BFS-2D.

Through the BEV-2D, we can intuitively verify the robustness of 4D Radars to adverse weather conditions as shown in Figure 2. As mentioned earlier, camera and Lidar measurements can deteriorate under adverse weather conditions such as rain, sleet, and snow. In Figure 2-(e,f), we show that the measurements of a Lidar for a long-distance object are lost in heavy snow conditions. However, the

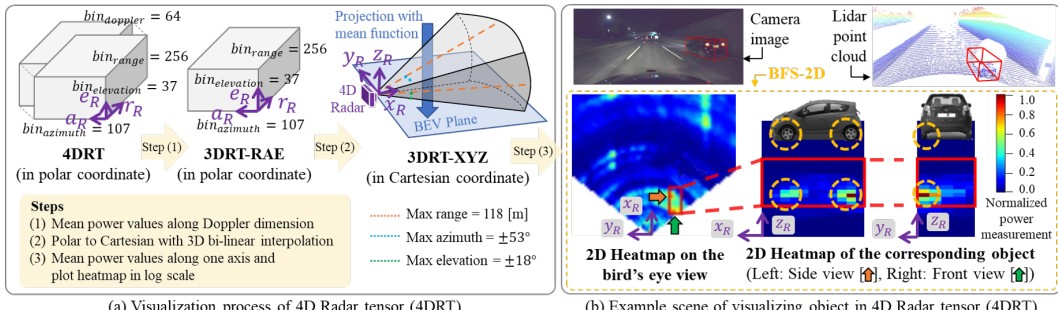

| (a) Visualization process of 4D Radar tensor (4DRT) | (b) Example scene of visualizing object in 4D Radar tensor (4DRT) |

Figure 6: (a) A 4DRT visualization process and (b) the 4DRT visualization results. (a) is the process of visualizing 4DRT (polar coordinate) into BFS-2D (Cartesian coordinate) through a three-step process: (1) extracting the 3D Radar tensor that contains measurements along the range, azimuth, and elevation dimensions (3DRT-RAE) by reducing the Doppler dimension of the 4DRT through dimension-wise averaging, (2) transforming the 3DRT-RAE (polar coordinate) into 3DRT-XYZ (Cartesian coordinate), (3) by removing one of the three dimensions of 3DRT-XYZ, the 4DRT is finally visualized as a two-dimensional Cartesian coordinate system. (b) is an example in which 4DRT-3D information is visualized as BFS-2D through the process of (a). We also show the front view camera image and the LPC of the same frame on the upper side of (b), and the bounding box of the car is marked in red. As shown in (b), the 4DRT is represented by three types of views (i.e., BEV, side view, and front view). We note that high power measurements are observed on wheels rather than the body of the vehicle when compared to the actual vehicle model picture with the side view and front view of the object. This is because radio wave reflection occurs mainly in wheels made of metal (Brisken et al., 2018), not in the body of a vehicle made of reinforced plastic.

BEV-2D of the 4DRT clearly indicate the object with high-power measurements on the edge of the bounding box of the objects.

Even with the BFS-2D, it is still challenging for a human annotator to recognize the shape of objects appearing on the frame and accurately annotate the corresponding 3D bounding boxes. Therefore, we create a tool that enables 3D bounding boxes annotation in LPCs where object shapes are more recognizable. In addition, we use the BEV-2D to help the annotators in the case of lost Lidar measurements due to adverse weather conditions. The details are covered in Appendix D.1.

We also present a tool for frame-by-frame calibration of the BEV-2D and the LPC to transform the 3D bounding box labels from the Lidar coordinate frame to the 4D Radar coordinate frame. The calibration tool supports a resolution of 1 cm per pixel with a maximum error of 0.5 cm. The details of calibration between 4D Radar and Lidar are covered in Appendix D.2.

Additionally, we precisely obtain the calibration parameter between Lidar and the camera through a series of processes detailed in Appendix D.3. The calibration process between Lidar and camera enables the 3D bounding boxes and LPCs to be projected accurately onto camera images, which is crucial for multi-modal sensor fusion study, and can be used to produce dense depth maps for monocular depth estimation study.

### 3.4 Baseline NNs for K-Radar

We provide two baseline NNs to demonstrate the importance of height information for 3D object detection: (1) Radar Tensor Network with Height (RTNH) that extracts feature maps (FMs) from RT with 3D Sparse CNN so that height information is utilized, and (2) Radar Tensor Network without Height (RTN) that extracts FMs from RT with 2D CNN that does not utilize height information.

As shown in Figure 7, both RTNH and RTN consist of pre-processing, backbone, neck, and head. The pre-processing transforms the 4DRT from polar to Cartesian coordinate frame and extracts a 3DRT-XYZ within the region of interest (RoI). Note that we reduce the Doppler dimension by taking the mean value along the dimension. The backbone then extracts FMs that contain important features for the bounding box predictions. And the head predicts 3D bounding boxes from the concatenated FM produced by the neck.

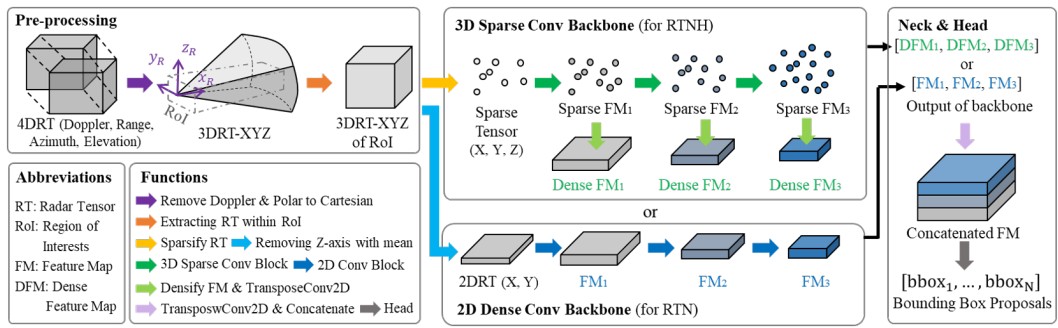

Figure 7: Two baseline NNs for verifying 4DRT-based 3D object detection performance.

The network structure of RTNH and RTN, described in details in Appendix E, is similar except the backbone. We construct the backbones of RTNH and RTN with 3D Sparse Conv Backbone (3D-SCB) and 2D Dense Conv Backbone (2D-DCB), respectively. 3D-SCB utilizes 3D sparse convolution (Liu et al., 2015) so that the three-dimensional spatial information (X, Y, Z) can be encoded into the final FM. We opt to use the sparse convolution on sparse RT (top-10% power measurements in the RT) since dense convolution on the original RT requires a prohibitively large amount of memory and computations that are unsuitable for real-time autonomous driving applications. Unlike 3D-SCB, 2D-DCB uses 2D convolution so that only two-dimensional spatial information (X, Y) is encoded into the final FM. As a result, the final FM produced by 3D-SCB contains 3D information (with height), whilst the final FM produced by 2D-DCB only contains 2D information (without height).

## 4   Experiment

In this section, we demonstrate the robustness of 4DRT-based perception for autonomous driving under various weathers in order to find 3D object detection performance comparison between the baseline NN and a similarly-structured Lidar-based NN, PointPillars (Lang et al., 2019). We also discuss the importance of height information by comparing 3D object detection performance between baseline NN with 3D-SCB backbone (RTNH) and baseline NN with 2D-DCB backbone (RTN).

### 4.1   Experiment Setup and Metric

**Implementation Detail** We implement the baseline NNs and PointPillars using PyTorch 1.11.0 on Ubuntu machines with a RTX3090 GPU. We set the batch size to 4 and train the networks for 11 epochs using Adam optimizer with a learning rate of 0.001. Note that we set the detection target to the sedan class, which has the largest number of samples in K-Radar dataset.

**Metric** In the experiments, we utilize the widely-used Intersection Over Union (IOU)-based Average Precision (AP) metric to evaluate the 3D object detection performance. We provide APs for BEV ($AP_{BEV}$) and 3D ($AP_{3D}$) bounding boxes predictions as in (Geiger et al., 2012), where a prediction is considered to be a true positive if the IoU is over 0.3.

Table 3: Performance comparison of baseline NNs with or without height Information.

| Baseline NNs | $AP_{3D}$ [%] | $AP_{BEV}$ [%] | GPU RAM [$MB$] |
|---|---|---|---|
| RTNH | **47.44** | **58.39** | **421** |
| RTN | 40.12 | 50.67 | 520 |

### 4.2   Comparison between RTN and RTNH

We show the detection performance comparison between RTNH and RTN on Table 3. We can observe that RTNH has 7.32% and 7.72% higher performance in $AP_{3D}$ and $AP_{BEV}$, respectively, compared to RTN. RTNH significantly surpasses RTN in terms of both $AP_{3D}$ and $AP_{BEV}$, indicating the importance of height information available in the 4DRT for 3D object detection. Furthermore, RTNH requires less GPU memory compared to RTN since it utilizes the memory-efficient sparse convolutions as mentioned in Section 3.4.

### 4.3 Comparison between RTNH and PointPillars

Table 4: Performance comparison of NNs of Radar and Lidar under various weather conditions

| Networks | Metric | Total | normal | overcast | fog | rain | sleet | light snow | heavy snow |
|---|---|---|---|---|---|---|---|---|---|
| RTNH | $AP_{3D}[\%]$ | **47.4** | 49.9 | **56.7** | **52.8** | 42.0 | **41.5** | **50.6** | **44.5** |
| (4D Radar) | $AP_{BEV}[\%]$ | **58.4** | **58.5** | **64.2** | **76.2** | **58.4** | **60.3** | **57.6** | **56.6** |
| PointPillars | $AP_{3D}[\%]$ | 45.4 | **52.3** | 56.0 | 42.2 | **44.5** | 22.7 | 40.6 | 29.7 |
| (Lidar) | $AP_{BEV}[\%]$ | 49.3 | 56.6 | 61.0 | 52.0 | 57.8 | 23.1 | 51.6 | 30.8 |

We show the detection performance comparison between RTNH and a similarly-structured Lidar-based detection network, PointPillars, in Table 4. The Lidar-based network suffers significant BEV and 3D detection performance drops of 33.5% and 29.6% or 25.8% and 22.6%, respectively, in sleet or heavy snow condition compared to the normal condition. In contrast, the 4D radar-based RTNH detection performance is hardly affected by adverse weathers, where the BEV and 3D object detection performances in sleet or heavy snow condition are better or similar compared to the normal condition. The results testify the robustness of 4D radar-based perception in adverse weathers. We provide qualitative results and additional discussions for other weather conditions in Appendix F.

## 5 Limitation and Conclusion

In this section, we discuss the limitations of K-Radar and provide a summary of this work, along with suggestions on the future research directions.

### 5.1 Limitation of the FOV coverage of 4DRTs

As mentioned in Section 3.1, K-Radar provides 4D radar measurements in the forward direction, with an FOV of 107 degree. The measurement coverage is more limited compared to the 360 degree FOV of Lidar and camera. This limitation is originated from the size of a 4DRT with dense measurements in four dimensions, which require significantly larger memory to store the data compared to a camera image with two dimensions or a LPC with three dimensions. Specifically, the size of the 4DRT data in K-Radar is roughly 12TB, while the size of surround camera images data is about 0.4TB, and the size of LPCs data is about 0.6TB. Since providing 360 degrees 4DRT measurements requires a prohibitively large amount of memory, we opt to record 4DRT data only in the forward direction, which could provide the most relevant information for autonomous driving.

### 5.2 Conclusion

In this paper, we have introduced a 4DRT-based 3D object detection dataset and benchmark, K-Radar. The K-Radar dataset consists of 35K frames with 4DRT, LPC, surround camera images, and RTK-IMU data, all of which are collected in various time and weather conditions. K-Radar provides 3D bounding box labels and tracking ID for 93.3K objects of five classes with distance of up to 120 m. To verify the robustness of 4D radar-based object detection, we introduce baseline NNs that uses 4DRT as the input. From experimental results, we demonstrate the importance of height information that is not available in the conventional 3DRT and the robustness of 4D radar under adverse weathers for 3D object detection. While the experiments in this work are focused on 4DRT-based 3D object detection, K-Radar can be used for 4DRT-based object tracking, SLAM, and various other perception tasks. Therefore, we hope that K-Radar can accelerate works in 4DRT-based perception for autonomous driving.

## Acknowledgment

This work was partly supported by Institute of Information & communications Technology Planning & Evaluation (IITP) grant funded by the Korea government (MSIT) (No. 01210790) and the National Research Foundation of Korea (NRF) grant funded by the Korea government (MSIT) (No. 2021R1A2C3008370).

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
