# OpenReview forum: "K-Radar: 4D Radar Object Detection for Autonomous Driving in Various Weather Conditions"
_NeurIPS.cc/2022/Track/Datasets_and_Benchmarks — NeurIPS 2022 Datasets and Benchmarks _

### Official Review · Reviewer_6HK3 · 2022-07-24
**Review of K-Radar**

**Rating:** 6
**Confidence:** 4
**Clarity:** Yes.

**Strengths:**

1. A 4D Radar dataset and benchmark for autonomous driving in various weather conditions.
2. This is the first dataset with 4DRT for Radar object detection.

**Weaknesses:**

1. The dataset has tracking IDs. However, the paper does not provide any experiments about tracking performance.
2. It lacks reference of the proposed baseline method. Even if the baseline method is proposed by authors, it is still built on existing works. Besides that, it should be better to adapt some existing methods to 4DRT input data for comparison.

**Additional Feedback:**

No

**Correctness:**

Most are correct. As mentioned in the weakness, the benchmark can be also extended to 3D tracking benchmark. The paper can also provide the evaluation for 3D tracking.

**Documentation:**

The paper provides details about the data collection and organization.

**Relation To Prior Work:**

Yes. It compares with previous datasets and benchmarks.

**Summary And Contributions:**

This paper proposes a 4D Radar object detection dataset and benchmark for autonomous driving in various weather conditions, including normal, sleet, rain, heavy snow, fog, light snow, and overcast. And the dataset is the first dataset with  4DRT, including Doppler, range, azimuth, and elevation. It provides the baseline method for 3D object detection.

---

> ### Author Response · Authors · 2022-08-20
> **Author Response to 6HK3**
>
> First of all, thank you so much for the review. We are encouraged by your positive rating score for our proposed dataset and the presented study.
>
> We have revised the paper with regards to the following issues based on your comments: __(1) Additional experiments for 3D object tracking, and (2) Referencing previous works for the baseline network__.

---

> > ### Author Response · Authors · 2022-08-20
> > **Response to (2) Referencing previous works for the baseline network**
> >
> > __Q2__. It lacks reference of the proposed baseline method. Even if the baseline method is proposed by authors, it is still built on existing works. Besides that, it should be better to adapt some existing methods to 4DRT input data for comparison.
> >
> > __A2__. Due to page limitations of the main paper, the details of the baseline network are described in Appendix E. We would like to note that references of each technique used in the baseline network are specified early in Appendix E (line 132\~137 and line 154\~156). For example, we specified (Simon et al.) for dividing the angle of rotation into two parameters in Appendix E.1, and (Deng et al.) for the concatenation of the Z-axis into the channel in Appendix E.2.
> >
> > (Simon et al.) Simony, Martin, et al. "Complex-yolo: An euler-region-proposal for real-time 3d object detection on point clouds." Proceedings of the European Conference on Computer Vision (ECCV) Workshops. 2018.
> >
> > (Deng et al.) Deng, Jiajun, et al. "Voxel r-cnn: Towards high performance voxel-based 3d object detection." Proceedings of the AAAI Conference on Artificial Intelligence. Vol. 35. No. 2. 2021.

---

> > ### Author Response · Authors · 2022-08-20
> > **Response to (1) Additional experiments for 3D object tracking**
> >
> > __Q1__. The dataset has tracking IDs. However, the paper does not provide any experiments about tracking performance.
> >
> > __A1__. To jumpstart the 4D Radar tensor-based deep learning research, we publish not only the labels for 3D object detection but also the labels (e.g., ID of each 3D bounding box, depth map, and position of ego-vehicle) for other tasks (e.g., multi-object tracking, depth estimation, deep-learning based odometry). For examples of tracking information and depth maps, please refer to Appendix K.7 (https://www.youtube.com/watch?v=0hid8flEwjs) and Appendix D.3, respectively. We plan to conduct research on 4D Radar tensor-based multi-object tracking and odometry for future work (based on the tracking information and depth maps).
> >
> > In addition, we explicitly refer to the Appendix in the main paper for the readers to clearly get the information regarding the tracking:
> >
> > In the line 88~89 of the revised main paper (copy and paste): “Examples of information regarding tracking are shown in Appendix K.7.”

---

### Official Review · Reviewer_fy92 · 2022-07-26
**K-Radar: 4D Radar Object Detection Dataset and Benchmark for Autonomous Driving in Various Weather Conditions**

**Rating:** 6
**Confidence:** 4
**Correctness:** Yes, the dataset is constructed in a …

**Strengths:**

1. The authors release the first public large-scale dataset for autonomous driving incorporating 4DRT data.
2. The data distribution is well organized over different collection times, weather conditions, and road types.
3. Experimental results show that height information is essential for the baseline 4DRT neural network (NN) and that performance of the baseline 4DRT NN under adverse weather conditions surpasses that of Lidar.

**Weaknesses:**

1. The authors do not mention the calibration process of other sensors to Lidar except radar. The yellow bounding boxes shown in https://youtu.be/b_9TtOxaN1w at 1:47 seem not aligned with cars on the image due to the camera and lidar calibration issue.

2. In section D.1 of the Appendix, annotators can verify the height and size information of the 3D bounding box with BFS-2D. However, the strong reflection regions of the side view on the heatmap correspond to vehicle wheels, as shown in Fig.6 (b). How do annotators verify the size information during annotation without the Lidar point cloud?

3. The authors used IoU over 0.3 as a true positive in their evaluation metric. Similar to other public datasets, results using stricter IoU criteria such as 0.5 or 0.7 should be reported.


**Additional Feedback:**

1. Could the authors also include the visualization of the 3D bounding box projected on the image planes during annotation? It would be helpful to verify the object class under most weather conditions.

2. In Fig.11, I wonder if the dash cam is calibrated with Lidar so that the bounding boxes shown on the dash cam video and the radar heat map are programmatically associated.


**Clarity:**

The paper is well written and provides sufficient supplementary material in the appendix.

**Documentation:**

There is sufficient detail on data collection and organization. Currently, the dataset is not released on the provided link yet.

**Ethics:**

No ethical concerns are found.

**Relation To Prior Work:**

Yes.

**Summary And Contributions:**

This paper presents the first large-scale dataset containing four-dimensional radar tensor (4DRT), Lidar,  stereo camera, IMU, and RTK-GPS data for the perception module in autonomous driving.
The authors collected and annotated the actual driving data under various weather conditions and road types, especially for adverse weather conditions such as heavy snow.
Additionally, they provide software for the dataset letting users calibrate Lidar and radar sensors, annotate the 3d bounding box and visualize annotation interactively.
Finally, they conducted two experiments to validate the importance of 4DRT in the perception module of autonomous driving.

---

> ### Author Response · Authors · 2022-08-20
> **Author Response to fy92**
>
> First of all, thank you so much for the review. We are encouraged by your positive rating score for our proposed dataset and the presented study.
>
> We have revised the paper with regards to the following issues based on your comments: __(1) Ambiguity of height information for labeling using the side view of BFS-2D, (2) Detection performance under stricter metric, (3) Feedback for the labeling program, (4) Question about the labeling program, and (5) Calibration between camera and Lidar__.

---

> > ### Author Response · Authors · 2022-08-20
> > **Response to (5) Calibration between camera and Lidar**
> >
> > __Q5__. The authors do not mention the calibration process of other sensors to Lidar except radar. The yellow bounding boxes shown in https://youtu.be/b_9TtOxaN1w at 1:47 seem not aligned with cars on the image due to the camera and lidar calibration issue.
> >
> > __A5__. Please refer to __[Response to (Issue #1) Calibration between camera and Lidar]__ in the __[Author response to the three common issues]__. Let us copy and paste our response below for your convenience:
> >
> > We agree that, in the original paper, the calibration between Lidar and 4D Radar is discussed in detail, but the calibration between Lidar and camera is not described enough. Therefore, in Appendix D.3 of the revision, we provide details of the full calibration procedure between Lidar and camera; from the lens distortion of the camera to the detailed calibration procedure and result between the Lidar and the camera.
> >
> > Summarizing the (added) calibration procedure in Appendix D.3, we first obtain coarse values of the intrinsic parameters (i.e., principal point, focal length, and lens distortion) and extrinsic parameters (i.e., rotation and translation between Lidar and camera) by using a checkerboard calibration provided in ROS [1] and 3D modeling of the sensor suite provided in iPhone 12 Pro [2], respectively. Then, we construct GUI (Graphical User Interfaces) to fine-tune these parameters, which enables us to reduce fine calibration errors (which [1] and [2] are difficult to guarantee) as seen in Figure 17 (added on page 8 of the revised Appendix). Finally, we show the accuracy of the calibration by illustrating the projected points onto the front image and generating depth maps as seen in Figure 18 and 19 (added on page 8 of the revised Appendix), respectively.
> >
> > Please note that the generated depth map is another contribution of our dataset; our dataset can be used as ground truth for camera-based depth estimation as well.
> >
> > [1] Stanford Artificial Intelligence Laboratory et al. Robotic operating system.
> >
> > [2] Gregor Luetzenburg, Aart Kroon, and Anders Bjørk. Evaluation of the apple iphone 12 pro lidar for an application in geosciences. Scientific Reports, 11, 2021.

---

> > ### Author Response · Authors · 2022-08-20
> > **Response to (4) Question about the labeling program**
> >
> > __Q4__. In Fig.11, I wonder if the dash cam is calibrated with Lidar so that the bounding boxes shown on the dash cam video and the radar heat map are programmatically associated.
> >
> > __A4__. The dash cam image is not programmatically associated with the radar heatmap because it does not contain a timestamp, unlike the front camera image. However, since the dash cam image is divided into one-minute videos, it is possible to find the dashcam associated with each sequence and can be utilized as a reference image in adverse weathers. Dash cam videos corresponding to each sequence are also published together.

---

> > ### Author Response · Authors · 2022-08-20
> > **Response to (3) Feedback for the labeling program**
> >
> > __Q3__. Could the authors also include the visualization of the 3D bounding box projected on the image planes during annotation? It would be helpful to verify the object class under most weather conditions.
> >
> > __A3__. We thank the reviewer for suggesting several improvements to the labeling program. As the reviewer mentioned, we have improved the labeling program so that the projected 3D bounding box can be seen on the front image. Please refer to Appendix K.7 (https://www.youtube.com/watch?v=0hid8flEwjs) for an example of 3D bounding box projected onto the image plane.

---

> > ### Author Response · Authors · 2022-08-20
> > **Response to (2) Detection performance under stricter metric**
> >
> > __Q1__. The authors used IoU over 0.3 as a true positive in their evaluation metric. Similar to other public datasets, results using stricter IoU criteria such as 0.5 or 0.7 should be reported.
> >
> > __A2__. The BEV and 3D detection performance for 0.5 IoU of RTNH is 41.18 and 14.98, respectively, and the BEV and 3D detection performance for 0.5 IoU of RTNH-Doppler in Appendix G is 50.53 and 22.92, respectively. We conduct this study to show the possibility of 4D Radar tensor-based 3D object detection with 4D radar as the primary sensor. Therefore, we use metric of 0.3 IoU, which can directly show the possibility of finding objects rather than evaluating precise localization. Of course, the development kits that we publish provide the function to easily set IoU thresholds so that researchers can evaluate whether 3D objects are accurately localized. We also look forward to the improvement of neural networks capable of precise localization, which show high performance even in strict evaluation of 0.5 or 0.7, through future works on this dataset.

---

> > ### Author Response · Authors · 2022-08-20
> > **Response to (1) Ambiguity of height information for labeling using the side view of BFS-2D**
> >
> > __Q1__. In section D.1 of the Appendix, annotators can verify the height and size information of the 3D bounding box with BFS-2D. However, the strong reflection regions of the side view on the heatmap correspond to vehicle wheels, as shown in Fig.6 (b). How do annotators verify the size information during annotation without the Lidar point cloud?
> >
> > __A1__. We appreciate your question on the important issues during labeling under adverse weather. As we mention early in Appendix D.1 and I.4, human annotators label objects by referring to both BFS-2D and dash cam when Lidar point cloud measurements are absent. The position of the wheels can be obtained from the BFS-2D, and the height of the vehicle is set to a pre-defined value, when the type of the vehicle is identified using the dash cam image. Therefore, the full 3D bounding box can be estimated even in the absence of Lidar measurements.
> >
> > To clearly state that the above explanation, we add a sentence in the line 78 ~ 79 of the revised Appendix as (copy and paste):
> >
> > “We note that the height of the vehicle is set to a pre-defined value, when the type of the vehicle is identified using the dash cam image.”

---

### Official Review · Reviewer_vuZA · 2022-07-27
**Good dataset but weak in benchmarking**

**Rating:** 8
**Confidence:** 5
**Correctness:** yes.
**Clarity:** yes.

**Strengths:**

- The paper is well-written. Figures and tables present the dataset clearly.
- Comparisons with other datasets are solid.
- The annotation/visualization tool could benefit future research or customized labels.

**Weaknesses:**

- One concern regarding privacy is if the vehicles and pedestrians in the dataset are processed with appropriate anonymization.
- It would be better if more methods are compared in the benchmarking. And discussions in failure patterns.


**Additional Feedback:**

please see the weakness section.

**Documentation:**

yes.

**Ethics:**

Not sure if the vehicles and pedestrians in the dataset are processed with appropriate anonymization.

**Relation To Prior Work:**

yes, it is clearly discussed.

**Summary And Contributions:**

- This paper presents a 4D Radar tensor (4DRT)-based dataset (K-Radar) for 3D object detection. It also provides diverse scenarios with challenging illumination, time, and weather conditions.
- A 3d detection baseline method is introduced and proves that 4DRT is essential for autonomous driving.

---

> ### Author Response · Authors · 2022-08-20
> **Author Response to vuZA**
>
> First of all, thank you so much for the review. We are encouraged by your high rating score for our proposed dataset and the presented study.
>
> We have revised the paper with regards to the following issues based on your comments: __(1) Privacy concerns, (2) Discussion in failure patterns, and (3) Additional baseline networks__.

---

> > ### Author Response · Authors · 2022-08-20
> > **Response to (3) Additional baseline networks**
> >
> > __Q3__. It would be better if more methods are compared in the benchmarking.
> >
> > __A3__. Please refer to __[Response to (Issue #2) Additional baseline networks]__ in the __[Author response to the three common comments]__. Let us copy and paste our response below for your convenience:
> >
> > We agree with your comment. In the revision, we provide additional baseline networks for 4D Radar (indicated by reviewer YMMm), Lidar (by reviewer 1XPW), and camera (by reviewer YMMm), which are summarized in the following.
> >
> > - __4D Radar__: As discussed in the revised Appendix G (Experiments for the baseline network utilizing Doppler measurements), Doppler measurements representing the relative radial velocities of surrounding objects are useful for object detection. Therefore, in the revision, we provide additional 4D Radar baseline network (i.e., RTNH-Doppler) that utilize Doppler measurements in addition to the XYZ. Note that since we focus on the elevation information that is available in 4DRT but not in the conventional 3DRT, the main body of the paper discusses RTNH. However, discussions for RTNH-Doppler is added to Appendix G, and we open the RTNH-Doppler code to the public as well.
> >
> > - __Lidar__: As in Section 4, we compare the baseline network for 4D Radar to PointPillars that is a well-known baseline network for Lidar, which is because these two networks have similar structures such as (1) one-stage detector and (2) the same neck and head. However, as in the review comments, PointPillars (2019) is not new. To improve the 3D detection performance comparison of the proposed Lidar baseline networks, we select a recent network, PV-RCNN++ (2021), that has two-stage detector structure and we provide the training codes as well. (The training on the K-Radar dataset is ongoing, and we will post it on the project page as soon as the training is complete.)
> >
> > - __Camera__: We are going to upload the training codes and 3D detection performance for a recent camera-based 3D object detection baseline network, CaDNN (2021). Since CaDNN requires dense depth maps for training, we are generating depth maps, which can take about 2 months. As soon as the generation of the dense depth map completed, we are planning to train the network and provide the detection performance on the project page.

---

> > ### Author Response · Authors · 2022-08-20
> > **Response to (2) Discussion in failure patterns**
> >
> > __Q2__. And discussions in failure patterns.
> >
> > __A2__. We mention early our failure patterns especially for the light snow conditions in Appendix F (line 192 ~ 197). As the road environments, which collects data in the light snow, is an environment with many metal structures (in the middle figure of Figure 22), which is measured with high power by the 4D Radar. Therefore, RTNH has a number of false alarms, resulting in less detection performance in the light snow conditions.

---

> > ### Author Response · Authors · 2022-08-20
> > **Response to (1) Privacy concerns**
> >
> > __Q1__. One concern regarding privacy is if the vehicles and pedestrians in the dataset are processed with appropriate anonymization.
> >
> > __A1__. We appreciate your concerns about the privacy of people in our dataset. We confirm that all the sequences with pedestrians, bicycles, and motorcycles do not have recognizable faces. Although everyone is wearing masks due to COVID-19 concerns, we have blurred all faces to protect their personal information. The example frames of blurred faces are shown in Figure 10 (added on page 4 of the revised Appendix) in Appendix A.5 (Privacy concerns).
> >
> > To clearly state that we have considered the privacy of people in our dataset, we added a sentence in the line 50 ~ 52 of the revised Appendix as (copy and paste):
> >
> > “We confirm that all the image sequences with pedestrians, bicycles, and motorcycles do not have recognizable faces. Although everyone is wearing a mask due to COVID-19, we have taken additional precautions and blurred all faces to protect their privacy as shown in Figure 10.”

---

### Official Review · Reviewer_zuKk · 2022-07-27
**A new radar dataset capable of training 3D detection model for autonomous driving**

**Rating:** 7
**Confidence:** 5

**Strengths:**

The proposed dataset contains raw radar tensor data which preserves more details that can be studied by future works to advance the detection performance in adverse weather conditions. They are the first work to provide raw radar tensor data with elevation information which is critical for 3D object detection. The proposed dataset, visualization tools, and annotation tools can bring impact to the research community.

**Weaknesses:**

The calibration process for the sensor suite is not very clear. The paper has mentioned the calibration between radar and lidar but did not explain other extrinsic calibrations for cameras. The camera extrinsic calibration is important as they show the dash-camera image as a supporting image for annotation in the appendix. The authors mentioned the calibrations between radar and lidar by adjusting the 2D translation and yaw angle, but the error of pitch angle introduced by the two sensors was not calibrated. The authors should consider elaborating more on the calibration process.

**Additional Feedback:**

I have a question about the performance of RTNH versus PointPillars under different weather conditions. Why the general AP of the normal condition is lower than the overcast condition? And why the RTNH bird's-eye view AP of the light-snow conditions is lower than heavy-snow conditions while 3D APs are pretty much the same? Perhaps authors can explain more about their experiment.

**Clarity:**

Overall, the paper was well written and organized. But I found most of the details about sensor calibration was in the appendix. The authors can consider move the sensor calibration details into the main page.

**Correctness:**

The paper claims to be a dataset paper and was constructed in a sound way. It has the introduction, related work, dataset collection and analysis. They also provided baseline with experiments to verify their dataset.

**Documentation:**

They are preparing to publish their full dataset. The documentation, development kit, and maintenance plan will be released together.

**Ethics:**

There is no ethics concern.

**Relation To Prior Work:**

The paper compared their dataset with other autonomous driving datasets and clearly state their advantages against others. The authors should consider including another radar object detection dataset RODNet [1] in their related works.

[1] Wang, Yizhou, et al. "RODNet: A real-time radar object detection network cross-supervised by camera-radar fused object 3D localization." IEEE Journal of Selected Topics in Signal Processing 15.4 (2021): 954-967.

**Summary And Contributions:**

In order to overcome the challenges of autonomous driving in adverse weather conditions, the authors collected and organized a new dataset with radar data. They introduce the radar sensor which is unaffected by poor illumination and capable of penetrating snow or rain drops due to its longer wavelength compared with the commonly used LiDAR sensor. The authors also introduced the signal processing for frequency modulated continuous wave radar signal. They carefully introduced their full sensor suite and the distribution of data collected under different weather conditions and lighting scenarios. The object’s type and distance distribution within the dataset has been analyzed in the paper. Visualization and annotation tools have also been introduced in the paper. The authors compare their dataset with other datasets designed for autonomous driving scenarios. In the end, they also provided a baseline neural network for 3D object detection trained using their dataset and compare the detection performance with the lidar detection model. The main contribution of this paper is the proposed dataset which enables 3D object detection using radar signals for autonomous driving research.

---

> ### Author Response · Authors · 2022-08-20
> **Author Response to zuKk**
>
> First of all, thank you so much for the review. We are encouraged by your high rating score for our proposed dataset and the presented study.
>
> We have revised the paper with regards to the following issues based on your comments: __(1) Calibration of the pitch angle between 4D Radar and Lidar, (2) Organization of the paper for the calibration part, (3) Questions about the detection performance, (4) Calibration between camera and Lidar, and (5) the missing related dataset__.

---

> > ### Author Response · Authors · 2022-08-20
> > **Response to (5) Missing dataset: CRUW**
> >
> > __Q5__. The paper compared their dataset with other autonomous driving datasets and clearly state their advantages against others. The authors should consider including another radar object detection dataset RODNet [1] in their related works.
> >
> > [1] Wang, Yizhou, et al. "RODNet: A real-time radar object detection network cross-supervised by camera-radar fused object 3D localization." IEEE Journal of Selected Topics in Signal Processing 15.4 (2021): 954-967.
> >
> > __A3__. Please refer to __[Response to (Issue #3) Missing datasets]__ in the __[Author response to the three common comments]__. Let us copy and paste our response below for your convenience:
> >
> > First of all, we thank the reviewers for notifying us of these datasets. In Table 1, and 2 of the revision, we add the missing related datasets, such as CRUW and VoD as follows.
> >
> > ***
> >
> > Table 1. Comparison of object detection datasets and benchmarks for autonomous driving. HR and LR refer to High Resolution Lidar with more than 64 channels and Low Resolution with less than 32 channels, respectively. Bbox., Tr.ID, and Odom. refer to bounding box annotation, tracking ID, and odometry, respectively. Bolded text indicates the best entry in each category.
> > | Dataset | Num.data | RT | RPC | LPC | Camera | GPS | Bbox. | Tr.ID | Odom. |
> > |:---:|:---:|:---:|:---:|:---:|:---:|:---:|:---:|:---:|:---:|
> > | K-Radar (ours) | 35K | **4D** | **4D** | **HR.** | __360.__ | __RTK__ | __3D__ | __O__ | __O__ |
> > | VoD | 8.7K | X | **4D** | **HR.** | Front | __RTK__ | __3D__ | __O__ | __O__ |
> > | Astyx | 0.5K | X | **4D** | LR. | Front | X | __3D__ | X | X |
> > | RADDet | 10K | 3D | 3D | X | Front | X | 2D | X | X |
> > | Zendar | 4.8K | 3D | 3D | LR. | Front | GPS | 2D | __O__ | O |
> > | RADIATE | 44K | 3D | 3D | LR. | Front | GPS | 2D | __O__ | O |
> > | CARRADA | 12.6K | 3D | 3D | X | Front | X | 2D | __O__ | X |
> > | CRUW | 396K | 3D | 3D | X | Front | X | Point | __O__ | X |
> > | NuScenes | 40K | X | 3D | LR. | __360.__ | __RTK__ | __3D__ | __O__ | __O__ |
> > | Waymo | 230K | X | X | **HR.** | z | X | __3D__ | __O__ | X |
> > | KITTI | 15K | X | X | **HR.** | Front | __RTK__ | __3D__ | __O__ | __O__ |
> > | BDD100k | **120M** | X | X | X | Front | __RTK__ | 2D | __O__ | __O__ |
> >
> > ***
> >
> > Table 2. Comparison of object detection datasets and benchmarks for autonomous driving. d/n refers to day and night. Bolded text indicates the best entry in each category.
> > | Dataset | Weather conditions | Time |
> > |:---:|:---:|:---:|
> > | K-Radar (ours) | **overcast, fog, rain, sleet, snow** | **d/n** |
> > | VoD | X | day |
> > | Astyx | X | day |
> > | RADDet | X | day |
> > | Zendar | X | day |
> > | RADIATE | overcast, fog, rain, snow | **d/n** |
> > | CARRADA | X | day |
> > | CRUW | X | day |
> > | NuScenes | overcast, rain | **d/n** |
> > | Waymo | overcast | **d/n** |
> > | KITTI | X | day |
> > | BDD100k | overcast, fog, rain, snow | **d/n** |
> >
> > ***
> >
> > In Section 2 of the revision (line 150 ~ 154), we also add discussion for CRUW as follows (copy and paste): “CRUW (Wang et al., 2021) provides a large number of 3DRTs, but annotations only provide 2D point locations of objects. VoD (Palffy et al., 2022) and Asytx (Meyer and Kuschk, 2019) provide 3D bounding box labels with 4DRPCs. However, the dense 4DRTs are not made available, and the number of samples in the datasets is relatively small (i.e., 8.7K and 0.5K frames).”

---

> > ### Author Response · Authors · 2022-08-20
> > **Response to (4) Calibration between camera and Lidar**
> >
> > __Q4__. The calibration process for the sensor suite is not very clear. The paper has mentioned the calibration between radar and lidar but did not explain other extrinsic calibrations for cameras. The camera extrinsic calibration is important as they show the dash-camera image as a supporting image for annotation in the appendix.
> >
> > __A4__. Please refer to __[Response to (Issue #1) Calibration between camera and Lidar]__ in the __[Author response to the three common issues]__. Let us copy and paste our response below for your convenience:
> >
> > We agree that, in the original paper, the calibration between Lidar and 4D Radar is discussed in detail, but the calibration between Lidar and camera is not described enough. Therefore, in Appendix D.3 of the revision, we provide details of the full calibration procedure between Lidar and camera; from the lens distortion of the camera to the detailed calibration procedure and result between the Lidar and the camera.
> >
> > Summarizing the (added) calibration procedure in Appendix D.3, we first obtain coarse values of the intrinsic parameters (i.e., principal point, focal length, and lens distortion) and extrinsic parameters (i.e., rotation and translation between Lidar and camera) by using a checkerboard calibration provided in ROS [1] and 3D modeling of the sensor suite provided in iPhone 12 Pro [2], respectively. Then, we construct GUI (Graphical User Interfaces) to fine-tune these parameters, which enables us to reduce fine calibration errors (which [1] and [2] are difficult to guarantee) as seen in Figure 17 (added on page 8 of the revised Appendix). Finally, we show the accuracy of the calibration by illustrating the projected points onto the front image and generating depth maps as seen in Figure 18 and 19 (added on page 8 of the revised Appendix), respectively.
> >
> > Please note that the generated depth map is another contribution of our dataset; our dataset can be used as ground truth for camera-based depth estimation as well.
> >
> > [1] Stanford Artificial Intelligence Laboratory et al. Robotic operating system.
> >
> > [2] Gregor Luetzenburg, Aart Kroon, and Anders Bjørk. Evaluation of the apple iphone 12 pro lidar for an application in geosciences. Scientific Reports, 11, 2021.

---

> > ### Author Response · Authors · 2022-08-20
> > **Response to (3) Questions about the detection performance**
> >
> > __Q3__. I have a question about the performance of RTNH versus PointPillars under different weather conditions. Why the general AP of the normal condition is lower than the overcast condition? And why the RTNH bird's-eye view AP of the light-snow conditions is lower than heavy-snow conditions while 3D APs are pretty much the same? Perhaps authors can explain more about their experiment.
> >
> > __A3__. The general AP of normal conditions can be lower than the overcast conditions, since the 4D Radar and Lidar are not prone to low lighting conditions. Furthermore, the road environment of the frames in the overcast condition only has two lanes on urban roads, whereas the road environment of the frames in the normal condition has many vehicles parked along the roadside in alleyways.
> >
> > As we mention early in Appendix F (line 192 ~ 197), the road environment, which collects data in the light snow, is an environment with many metal structures, including guide rails in the center of the road. These metal structures are measured with high reflection power by the 4D Radar, and may increase the number of false alarms where they are incorrectly recognized as vehicles, which results in less detection performance in light snow conditions.
> >
> > In the line 180 ~ 187 of the revised Appendix, we add additional discussion for the readers to understand the performance of RTNH vs. PointPillars under various conditions as (copy and paste below):
> >
> > “In addition, notice that the general AP for the normal condition can be lower than the overcast condition on K-Radar because of the following two reasons. One reason is that 4D Radar and Lidar are not affected by lighting condition, so that the detection performance of 4D Radar and Lidar for overcast condition cannot be lower than that for normal condition. Another reason is that the normal condition in K-Radar has more difficult situation than the overcast condition; the normal condition contains various situations including many vehicles parked along the side of alleyways, while the overcast condition in K-Radar does not have many vehicles on clear urban roads of two lanes, as shown in Table 5.”

---

> > ### Author Response · Authors · 2022-08-20
> > **Response to (2) Organization of the paper for the calibration part**
> >
> > __Q2__. Overall, the paper was well written and organized. But I found most of the details about sensor calibration was in the appendix. The authors can consider move the sensor calibration details into the main page.
> >
> > __A2__. Due to the limited page count of the main paper, we have moved the majority of the calibration section to the Appendix. Please understand. However, we agree that the calibration between sensors is one of the most important parts of this paper, so we explicitly refer to the Appendix in the main paper for the readers to clearly get the information:
> >
> > In the line 230 ~ 231 of the revised main paper, we add an explanation as (copy and paste): “The details of calibration between 4D Radar and Lidar are covered in Appendix D.2.”

---

> > ### Author Response · Authors · 2022-08-20
> > **Response to (1) Calibration of the pitch angle between 4D Radar and Lidar**
> >
> > __Q1__. The authors mentioned the calibrations between radar and lidar by adjusting the 2D translation and yaw angle, but the error of pitch angle introduced by the two sensors was not calibrated. The authors should consider elaborating more on the calibration process.
> >
> > __A1__. We have not considered the pitch angle difference between 4D Radar and Lidar, since we fix the sensors precisely perpendicular to the ground, resulting in no theoretical difference in the pitch angle. Although we have not considered the pitch angle, the measurements for the far objects (e.g., Appendix K.5 around 3:53) match, which would not be possible in the presence of the difference in the pitch angle.
> >
> > To clearly state that there is no difference in the pitch angle between the Radar and Lidar, we add a statement in the line 89 ~ 91 of the revised Appendix as (copy and paste):
> >
> > “We have not considered the pitch angle difference between 4D Radar and Lidar, since we fix the sensors precisely perpendicular to the ground, resulting in no theoretical difference in the pitch angle.”

---

### Official Review · Reviewer_1XPW · 2022-07-28

**Rating:** 6
**Confidence:** 4
**Clarity:** Yes

**Strengths:**

- The new dataset provides 3D bounding box labels and tracking ID for 93.3K objects of five classes.
- The new dataset also provides boxes in diverse conditions and times.
- Unlike conventional 3DRT, K-radar contains power measurements along the Doppler, range, azimuth, and elevation dimensions, enabling precise 3D object detection.
- The paper shows that height information is crucial for 3D object detection.
- The 4DRT-based baseline outperforms the old lidar-based baseline in the 3D object detection task, especially in adverse weather conditions.
- Excellent documentation

**Weaknesses:**

I am willing to raise my score if the authors provide response to my questions.

- Table 1 in the paper does not quantify the statistics of existing datasets. Please quantitatively compare the datasets keeping the following columns in the table.

    - #Images
    - #Labelled Images
    - #3D boxes
    - #Train images
    - #Val images
    - #Test images
    - #Drive (hours)
    - #Area covered (km^2)
    - #Cities
    - Multiple Times of the day (Yes/No)
    - Multiple Lanes (Yes/No)
    - Hilly regions (Yes/No)
    - Videos (Yes/No)
    - Rainy images (Yes/No)
    - Snowy images (Yes/No)
    - Foggy images (Yes/No)

- Most bigger benchmarks compare the object detection performance over multiple classes. This paper compares the detection performance of the sedan class (L257). Please quantitatively report the performance of the baseline and proposed model on all the five classes rather than the sedan class.

- The pointpillars is a pretty old baseline for lidar-based object detection. Why did the authors choose this baseline? Please quantitatively report the results of using newer baselines from 2021.

- Since other datasets do not have the height in the radar modality, it would be good to quantitatively compare K-Radar and Radiant as a pre-training dataset. If the K-Radar is indeed diverse, we should see a strong improvement after training with the K-Radar dataset.

- Next, quantitatively report the performance of the model trained on K-Radar without height and evaluated on RADIATE. Similary, quantitavely report the performance of RADIATE dataset trained model on K-Radar (without height).


**Additional Feedback:**

NA

**Correctness:**

- Most bigger benchmarks compare the object detection performance over multiple classes. This paper compares the detection performance of the sedan class (L257). Please quantitatively report the performance of the baseline and proposed model on all the five classes rather than the sedan class.

- The pointpillars is a pretty old baseline for lidar-based object detection.One can not use an old baseline to claim that 4DRT-based baseline outperforms lidar based methods.

**Documentation:**

Excellent documentation

**Relation To Prior Work:**

Not completely. The authors did describe the prior work, but do not quantify the statistics of existing datasets.

**Summary And Contributions:**

The paper introduces a 4DRT-based 3D object detection dataset and benchmark, K-Radar. Unlike conventional 3DRT, K-radar contains power measurements along the Doppler, range, azimuth, and elevation dimensions. The paper also shows the baseline results on this dataset for the 3D object detection task. Moreover, the paper shows that height information is crucial for 3D object detection.

---

> ### Author Response · Authors · 2022-08-20
> **Author Response to 1XPW**
>
> First of all, thank you so much for the review and comments to improve the quality of the paper.
>
> We have revised the paper with regards to the following issues based on your comments: __(1) Additional information for the related dataset, (2) Experiment results for the multi-class detection, (3) Utilizing another 3D Radar tensor dataset, RADIATE, and (4) Additional baseline networks for Lidar__.

---

> > ### Author Response · Authors · 2022-08-20
> > **Response to (4) Additional baseline networks for Lidar**
> >
> > __Q4__. The pointpillars is a pretty old baseline for lidar-based object detection. Why did the authors choose this baseline? Please quantitatively report the results of using newer baselines from 2021.
> >
> > __A4__. Please refer to __[Response to (Issue #2) Additional baseline networks]__ in the __[Author response to the three common comments]__. Let us copy and paste our response below for your convenience:
> >
> > We agree with your comment. In the revision, we provide additional baseline networks for 4D Radar (indicated by reviewer YMMm), Lidar (by reviewer 1XPW), and camera (by reviewer YMMm), which are summarized in the following.
> >
> > - __4D Radar__: As discussed in the revised Appendix G (Experiments for the baseline network utilizing Doppler measurements), Doppler measurements representing the relative radial velocities of surrounding objects are useful for object detection. Therefore, in the revision, we provide additional 4D Radar baseline network (i.e., RTNH-Doppler) that utilize Doppler measurements in addition to the XYZ. Note that since we focus on the elevation information that is available in 4DRT but not in the conventional 3DRT, the main body of the paper discusses RTNH. However, discussions for RTNH-Doppler is added to Appendix G, and we open the RTNH-Doppler code to the public as well.
> >
> > - __Lidar__: As in Section 4, we compare the baseline network for 4D Radar to PointPillars that is a well-known baseline network for Lidar, which is because these two networks have similar structures such as (1) one-stage detector and (2) the same neck and head. However, as in the review comments, PointPillars (2019) is not new. To improve the 3D detection performance comparison of the proposed Lidar baseline networks, we select a recent network, PV-RCNN++ (2021), that has two-stage detector structure and we provide the training codes as well. (The training on the K-Radar dataset is ongoing, and we will post it on the project page as soon as the training is complete.)
> >
> > - __Camera__: We are going to upload the training codes and 3D detection performance for a recent camera-based 3D object detection baseline network, CaDNN (2021). Since CaDNN requires dense depth maps for training, we are generating depth maps, which can take about 2 months. As soon as the generation of the dense depth map completed, we are planning to train the network and provide the detection performance on the project page.

---

> > ### Author Response · Authors · 2022-08-20
> > **Response to (3) utilizing another 3D Radar tensor dataset, RADIATE**
> >
> > __Q3__. Since other datasets do not have the height in the radar modality, it would be good to quantitatively compare K-Radar and Radiant as a pre-training dataset. If the K-Radar is indeed diverse, we should see a strong improvement after training with the K-Radar dataset. Next, quantitatively report the performance of the model trained on K-Radar without height and evaluated on RADIATE. Similary, quantitavely report the performance of RADIATE dataset trained model on K-Radar (without height).
> >
> > __A3__. This maybe an interesting experiment to see. Thanks for the comment. However, we would like to emphasize two key aspects in our response to the comment; __(1) the data distribution is way different between K-Radar and RADIATE__, and __(2) the goal of the K-Radar dataset and the paper is to show the improvement in 3D object detection performance but not at all in 2D object detection performance__. Let us continue our answer in more detail about the above two aspects regarding the comment.

---

> > > ### Author Response · Authors · 2022-08-20
> > > **(2) Purpose of K-Radar**
> > >
> > > As stated above, the goal of the K-Radar dataset and the paper is to show the improvement in 3D object detection performance but not at all in 2D object detection performance. Therefore, we believe that quantitatively comparing the 2D detection performance of a network trained on K-Radar without the height to another network trained on RADIATE does not bring useful knowledge to readers.

---

> > > ### Author Response · Authors · 2022-08-20
> > > **(1) Data distribution difference**
> > >
> > > We want to note that the diversity of K-Radar does not directly guarantee a strong improvement in RADIATE as indicated by the reviewer. This is because the characteristics of K-Radar and RADIATE are inherently different in many ways.
> > >
> > > First, the power measurements in K-Radar and RADIATE have different distributions due to the different type of radars used. The inconsistent data distributions often lead to a poor performance of neural network in the new domain, as we see in Lidar object detection networks trained and evaluated on different type of point clouds (e.g., Velodyne and Ouster).
> > >
> > > Second, the resolution of RADIATE (0.175m) is higher than K-Radar (0.46m). The mismatched resolutions can also adversely affect the detection performance as usually seen in Lidar object detection networks trained on NuScenes (32-channels) and evaluated on KITTI (64-channels).
> > >
> > > Third, the data distributions can be significantly different. K-Radar data is collected in South Korea where cars drive on the right, while RADIATE is collected in the U.K where cars drive on the left.
> > >
> > > We refer to [1] which shows that 3D object detection performance degrades when the training and evaluation dataset are different.
> > >
> > > [1] Wang, Yan, et al. "Train in Germany, test in the USA: Making 3d object detectors generalize." Proceedings of the IEEE/CVF Conference on Computer Vision and Pattern Recognition. 2020.
> > >
> > > As a conclusion, we agree that this is an interesting viewpoint and it might be useful to discuss the above difference in the paper. We are adding the above explanation in the revised Appendix I (line 234 ~ 254) that is copy and pasted as below:
> > >
> > > “Pre-training on large scale datasets is well known to help neural networks converge faster (He et al., 2019). Therefore, we may consider using K-Radar as a pre-training dataset for other Radar tensor-based object detection datasets, or conversely, using other datasets as a pre-training dataset for K-Radar.
> > >
> > > However, we want to note that the pre-training on K-Radar does not directly guarantee a strong improvement on RADIATE. This is because the characteristics of K-Radar and RADIATE are inherently different.
> > >
> > > First, the power measurements in K-Radar and RADIATE have different distributions due to the different type of Radars used. When a neural network is trained on a dataset and applied to process target data of different distribution, there will be a poorly degraded performance in the target domain, as we see in Lidar object detection networks trained and evaluated on different type of point clouds (e.g., Velodyne and Ouster) (Wang et al., 2020).
> > >
> > > Second, the resolution of RADIATE (0.175m) is higher than K-Radar (0.46m). This mismatch of resolution can also adversely affect the detection performance as usually seen in Lidar object detection networks trained on NuScenes (32-channels) and evaluated on KITTI (64-channels) (Wang et al., 2020).
> > >
> > > Third, the data distributions are significantly different. K-Radar data is collected in South Korea where cars drive on the right, while RADIATE is collected in the U.K where cars drive on the left.
> > >
> > > The above reasons apply to other Radar tensor-based datasets as well as RADIATE. For these reasons, it is difficult to expect performance improvement by using K-Radar as a pre-training dataset for other Radar tensor-based datasets and vice versa.”
> > >
> > > (He et al., 2019) He, Kaiming, Ross Girshick, and Piotr Dollár. "Rethinking imagenet pre-training." Proceedings of the IEEE/CVF International Conference on Computer Vision. 2019.
> > >
> > > (Wang et al., 2020) Wang, Yan, et al. "Train in germany, test in the usa: Making 3d object detectors generalize." Proceedings of the IEEE/CVF Conference on Computer Vision and Pattern Recognition. 2020.”

---

> > ### Author Response · Authors · 2022-08-20
> > **Response to (2) Experiment results for the multi-class detection**
> >
> > __Q2__. Most bigger benchmarks compare the object detection performance over multiple classes. This paper compares the detection performance of the sedan class (L257). Please quantitatively report the performance of the baseline and proposed model on all the five classes rather than the sedan class.
> >
> > __A2__. Thank you for the comment that we didn’t consider thoroughly. We provide additional experiment results for multi-class detection in Appendix H (Experiments for multi-class detection). We consider sedans (small-size vehicles), buses (large-size vehicles), and pedestrians, which occupy the first, second, and third largest distributions because they have at least more than 5% of the total bounding boxes, respectively. We have not yet considered the bicycle and the motorcycle, not just because these classes occupy only 1% and 0.4% of the bounding boxes, but also because they are not enough to train neural networks, respectively.
> >
> > In the line 229 ~ 231 of the revised Appendix, we add an additional note for the readers regarding this issue as (copy and paste below):
> >
> > “We note that the detection performance for bicycles and motorcycles is not considered in Table 11, since these classes are rarely observed in the K-Radar dataset. In fact, bicycle and motorcycle occupy only 1% and 0.4% of the bounding boxes, respectively.”
> >
> > To consider the detection of these long-tails cases, we plan to investigate advanced training techniques as a future work.

---

> > ### Author Response · Authors · 2022-08-20
> > **Response to (1) Additional information for the related dataset**
> >
> > __Q1__. Table 1 in the paper does not quantify the statistics of existing datasets. Please quantitatively compare the datasets keeping the following columns in the table. (#Images, #Labelled Images, #3D boxes, #Train images, #Val images, #Test images, #Drive (hours), #Area covered (km^2), #Cities, Multiple Times of the day (Yes/No), Multiple Lanes (Yes/No), Hilly regions (Yes/No), Videos (Yes/No), Rainy images (Yes/No), Snowy images (Yes/No), Foggy images (Yes/No))
> >
> > __A1__. Thank you for this comment. We agree that additional statistics of the existing datasets can be helpful for the readers to comprehensively compare the pros and cons of the datasets.
> > We note that "data" includes camera images, Lidar point clouds, and 4D Radar tensors, so we use the term "data" instead of the term "image" used by the reviewer.
> >
> > As shown in Table A (please refer to the comment below), we have investigated statistics based on the information reported in other dataset papers and K-Radar. As stated in Table A, since not all datasets specify the items suggested by the reviewer, there are many items that are not available. In particular, to clearly understand the total number of datasets each dataset provides, one should download complete datasets and count the number of frames in each data. Because this requires a large hard disk capacity and time, and the internet bandwidth is different for each server that provides each dataset, the number of frames in each data is marked as 'n/a' when it is not mentioned in the paper.
> >
> > In addition, "#Area covered (km^2)", "#Cities", "Multiple Times of the day", "Multiple Lanes", "Hilly regions", and "Videos" are information that are ambiguous definitions or not important to show the diversity of distributions in datasets. Let us provide some explanation.
> >
> > For example, "# Area covered (km2)" is ambiguous; whether it refers to the area of the road from which the data is collected or the area over a region in the city. In addition, in the case of "Videos", it is ambiguous whether it means collection of consecutive camera images or videos containing camera images in mp4 format. When it comes to "Videos", our dataset distributes all camera images corresponding to 30 frames per second, similar to BDD100k. "Multiple Lanes" can be ambiguous too; whether there are multiple lanes on the road or multiple lane labels available. When it comes to "Multiple lanes", our dataset contains eight round-trip lanes of data.
> >
> > On the other hand, "#Cities" and "Multiple Times of the day" maybe ambiguous and not appropriate to show the diversity of distribution in datasets. For instance, even if one measures data in many cities, if the measurement is only on roads like urban roads, the distribution of datasets is not diverse because the types of roads are almost the same. Conversely, if one measures data in a single city on different types of roads, such as urban roads, alleyways, and parking lots, the distribution of datasets can be diverse. For the K-Radar dataset, the dataset is obtained from roads such as urban roads, highways, alleyways, suburban roads, university, mountain (sloped or hilly) roads, parking lots, and shoulders, as specified in Table A and Table 5 of Appendix A.
> >
> > Similarly, even if you measure data over several days, the distribution of datasets may not vary a lot, if you measure data only on daylight and sunny days. On the contrary, even if you measure data only within a few days, you can have diverse distribution of the dataset if you measure data in day and night under different weather conditions. For the K-Radar dataset, as mentioned in Table A and Table 2 in the main paper, data are obtained for day/night and under various weather conditions.
> >
> > Lastly, please refer to Table A or Table 2 in the main paper for "Rainy data", "Snowy data", and "Foggy data".
> >
> > However, due to space limitations (e.g., limitation of the width in pdf file), it is hard to show all of the statistics in a single table in the paper. Therefore, we provide the statistics in two separate tables (Table 1 and Table 2) in the main paper. Additionally, we also provide the combined tables including Table A on our project page so the readers can easily compare all characteristics of the datasets in a single table.

---

> > > ### Author Response · Authors · 2022-08-20
> > > **Table A. Additional information of K-Radar and related datasets.**
> > >
> > > Since the width of the current page is not large enough to show all the items of Table A, we show Table A divided into Table A (#1) and Table A (#2). Please refer to this link (https://github.com/kaist-avelab/K-Radar/blob/main/docs/related_datasets.md) for the combined table that helps the readers easily compare all the characteristics of the datasets.
> > >
> > > ***
> > >
> > > Table A (#1). Additional information of K-Radar and related datasets. HR and LR refer to High Resolution Lidar with more than 64 channels and Low Resolution with less than 32 channels, respectively. Bbox., Tr.ID, and Odom. refer to bounding box annotation, tracking ID, and odometry, respectively. ‘-’, and ‘n/a’ denote ‘not relevant’ and ‘not applicable’, respectively.
> > >
> > > | Dataset | Radar tensor | Radar point cloud | Lidar point cloud | Camera | GPS | Bbox label | Tr. ID | Odom. | Weather conditions | Time | Num. labelled data | Num. labelled train data | Num. labelled val. data | Num. labelled test data |
> > > |:---:|:---:|:---:|:---:|:---:|:---:|:---:|:---:|:---:|:---:|:---:|:---:|:---:|:---:|:---:|
> > > | K-Radar (ours) | 4D | 4D | HR. | 360. | RTK | 3D | O | O | overcast, fog, rain, sleet, snow | d/n | 35K | 17.5K | - | 17.5K |
> > > | VoD | X | 4D | HR. | Front | RTK | 3D | O | O | X | day | 8.7K | 5.1K | 1.3K | 2.3K |
> > > | Astyx | X | 4D | LR. | Front | X | 3D | X | X | X | day | 0.5K | 0.4K | - | 0.1K |
> > > | RADDet | 3D | 3D | X | Front | X | 2D | X | X | X | day | 10K | 8K | - | 2K |
> > > | Zendar | 3D | 3D | LR. | Front | GPS | 2D | O | O | X | day | 4.8K | n/a | - | n/a |
> > > | RADIATE | 3D | 3D | LR. | Front | GPS | 2D | O | O | overcast, fog, rain, snow | d/n | 44K | 33K | - | 11K |
> > > | CARRADA | 3D | 3D | X | Front | X | 2D | O | X | X | day | n/a | n/a | n/a | n/a |
> > > | CRUW | 3D | 3D | X | Front | X | Point | O | X | X | day | n/a | n/a | n/a | n/a |
> > > | NuScenes | X | 3D | LR. | 360. | RTK | 3D | O | O | overcast, rain | d/n | 40K | 28K | 6K | 6K |
> > > | Waymo | X | X | HR. | 360. | X | 3D | O | X | overcast | d/n | 230K | 160K | 40K | 30K |
> > > | KITTI | X | X | HR. | Front | RTK | 3D | O | O | X | day | 15K | 7.5K | - | 7.5K |
> > > | BDD100k | X | X | X | Front | RTK | 2D | O | O | overcast, fog, rain, snow | d/n | n/a | n/a | n/a | n/a |
> > >
> > > ***
> > >
> > > Table A (#2). Additional information of K-Radar and related datasets. HR and LR refer to High Resolution Lidar with more than 64 channels and Low Resolution with less than 32 channels, respectively. Bbox., Tr.ID, and Odom. refer to bounding box annotation, tracking ID, and odometry, respectively. ‘-’, and ‘n/a’ denote ‘not relevant’ and ‘not applicable’, respectively.
> > >
> > > | Dataset | Num. Radar data | Num. Lidar data | Num. camera data | Num. 3D bboxes | Num. 2D bboxes | Num. points of objects | Road type | Driving period [hour] | Maximum range of Radar [m] |
> > > |:---:|:---:|:---:|:---:|:---:|:---:|:---:|:---:|:---:|:---:|
> > > | K-Radar (ours) | 38.9K | 37.7K | 112K | 93K | - | - | urban, highway, alleyway, suburban, university, mountain, parkinglots, shoulder | 1 | 118 |
> > > | VoD | n/a | n/a | n/a | 123K | - | - | urban | 0.2 | 64 |
> > > | Astyx | n/a | n/a | n/a | 3K | - | - | urban | 0.01 | 100 |
> > > | RADDet | n/a | n/a | n/a | - | 2.8K | - | n/a | n/a | 50 |
> > > | Zendar | n/a | n/a | n/a | - | 11.3K | - | urban | n/a | 90 |
> > > | RADIATE | n/a | n/a | n/a | - | 200K | - | urban, highway, parkinglots, suburban | 3 | 100 |
> > > | CARRADA | n/a | n/a | n/a | - | 7K | - | parkinglots | 0.35 | 50 |
> > > | CRUW | 396K | - | n/a | - | - | 260K | urban, highway, parkinglots, suburban | 3.5 | n/a |
> > > | NuScenes | n/a | n/a | n/a | 1.4M | - | - | urban, highway suburban | 5.5 | n/a |
> > > | Waymo | - | n/a | n/a | 12M | - | - | urban, suburban | 6.4 | - |
> > > | KITTI | - | n/a | n/a | 80K | - | - | suburban, highway | 1.5 | - |
> > > | BDD100k | - | - | 120M | - | 3.3M | - | urban, highway, parkinglots | 1.1K | - |
> > >
> > > ***

---

### Official Review · Reviewer_YMMm · 2022-07-28
**Good dataset for autonomous driving applications**

**Rating:** 7
**Confidence:** 5
**Correctness:** The dataset construction is sound.

**Strengths:**

- Scale of the dataset is good with a variety of different weather conditions.
- The radar sensor they use is advanced with 4D tensor representation for many radar perception tasks.

**Weaknesses:**

- Baseline and benchmarking are not sufficient. More radar-based baselines can be added.
- Some other baselines are also recommended to be added like camera-based methods.
- Some related dataset comparison is missing (see below).

**Additional Feedback:**

N/A

**Clarity:**

Paper writing is clear. However, the calibration part should be more detailed described, especially for camera and lidar extrinsics calibration.

**Documentation:**

Good documentation is provided.

**Relation To Prior Work:**

Missing related dataset:
[1] Wang, Yizhou, et al. "Rethinking of Radar's Role: A Camera-Radar Dataset and Systematic Annotator via Coordinate Alignment." Proceedings of the IEEE/CVF Conference on Computer Vision and Pattern Recognition. 2021.

**Summary And Contributions:**

This paper introduces a new dataset called K-Radar. It includes three common sensors for autonomous driving applications, i.e., camera, radar, and lidar. The radar they are using is more advanced than the hardware in most existing datasets, which is very impressive. Besides, the variety of weather conditions is also very competitive. Although some limitations exist, it would still be a good dataset and helpful for related research.

---

> ### Author Response · Authors · 2022-08-20
> **Author Response to YMMm**
>
> First of all, thank you so much for the review. We are encouraged by your high rating score for our proposed dataset and the presented study.
>
> We have revised the paper with regards to the following issues based on your comments: __(1) Calibration between camera and Lidar, (2) Additional baseline networks for 4D Radar and camera, and (3) Missing dataset: CRUW__.

---

> > ### Author Response · Authors · 2022-08-20
> > **Response to (3) Missing dataset: CRUW**
> >
> > __Q3__. Missing related dataset: [1] Wang, Yizhou, et al. "Rethinking of Radar's Role: A Camera-Radar Dataset and Systematic Annotator via Coordinate Alignment." Proceedings of the IEEE/CVF Conference on Computer Vision and Pattern Recognition. 2021
> >
> > __A3__. Please refer to __[Response to (Common Issue #3) Missing datasets]__ in the __[Author response to the three common comments]__. Let us copy and paste our response below for your convenience:
> >
> > First of all, we thank the reviewers for notifying us of these datasets. In Table 1, and 2 of the revision, we add the missing related datasets, such as CRUW and VoD as follows.
> >
> > ***
> >
> > Table 1. Comparison of object detection datasets and benchmarks for autonomous driving. HR and LR refer to High Resolution Lidar with more than 64 channels and Low Resolution with less than 32 channels, respectively. Bbox., Tr.ID, and Odom. refer to bounding box annotation, tracking ID, and odometry, respectively. Bolded text indicates the best entry in each category.
> > | Dataset | Num.data | RT | RPC | LPC | Camera | GPS | Bbox. | Tr.ID | Odom. |
> > |:---:|:---:|:---:|:---:|:---:|:---:|:---:|:---:|:---:|:---:|
> > | K-Radar (ours) | 35K | **4D** | **4D** | **HR.** | __360.__ | __RTK__ | __3D__ | __O__ | __O__ |
> > | VoD | 8.7K | X | **4D** | **HR.** | Front | __RTK__ | __3D__ | __O__ | __O__ |
> > | Astyx | 0.5K | X | **4D** | LR. | Front | X | __3D__ | X | X |
> > | RADDet | 10K | 3D | 3D | X | Front | X | 2D | X | X |
> > | Zendar | 4.8K | 3D | 3D | LR. | Front | GPS | 2D | __O__ | O |
> > | RADIATE | 44K | 3D | 3D | LR. | Front | GPS | 2D | __O__ | O |
> > | CARRADA | 12.6K | 3D | 3D | X | Front | X | 2D | __O__ | X |
> > | CRUW | 396K | 3D | 3D | X | Front | X | Point | __O__ | X |
> > | NuScenes | 40K | X | 3D | LR. | __360.__ | __RTK__ | __3D__ | __O__ | __O__ |
> > | Waymo | 230K | X | X | **HR.** | z | X | __3D__ | __O__ | X |
> > | KITTI | 15K | X | X | **HR.** | Front | __RTK__ | __3D__ | __O__ | __O__ |
> > | BDD100k | **120M** | X | X | X | Front | __RTK__ | 2D | __O__ | __O__ |
> >
> > ***
> >
> > Table 2. Comparison of object detection datasets and benchmarks for autonomous driving. d/n refers to day and night. Bolded text indicates the best entry in each category.
> > | Dataset | Weather conditions | Time |
> > |:---:|:---:|:---:|
> > | K-Radar (ours) | **overcast, fog, rain, sleet, snow** | **d/n** |
> > | VoD | X | day |
> > | Astyx | X | day |
> > | RADDet | X | day |
> > | Zendar | X | day |
> > | RADIATE | overcast, fog, rain, snow | **d/n** |
> > | CARRADA | X | day |
> > | CRUW | X | day |
> > | NuScenes | overcast, rain | **d/n** |
> > | Waymo | overcast | **d/n** |
> > | KITTI | X | day |
> > | BDD100k | overcast, fog, rain, snow | **d/n** |
> >
> > ***
> >
> > In Section 2 of the revision (line 150 ~ 154), we also add discussion for CRUW as follows (copy and paste): “CRUW (Wang et al., 2021) provides a large number of 3DRTs, but annotations only provide 2D point locations of objects. VoD (Palffy et al., 2022) and Asytx (Meyer and Kuschk, 2019) provide 3D bounding box labels with 4DRPCs. However, the dense 4DRTs are not made available, and the number of samples in the datasets is relatively small (i.e., 8.7K and 0.5K frames).”

---

> > ### Author Response · Authors · 2022-08-20
> > **Response to (2) Additional baseline networks for 4D Radar and camera**
> >
> > __Q2__. Baseline and benchmarking are not sufficient. More radar-based baselines can be added. & Some other baselines are also recommended to be added like camera-based methods.
> >
> > __A2__. Please refer to __[Response to (Common Issue #2) Additional baseline networks]__ in the __[Author response to the three common comments]__. Let us copy and paste our response below for your convenience:
> >
> > We agree with your comment. In the revision, we provide additional baseline networks for 4D Radar (indicated by reviewer YMMm), Lidar (by reviewer 1XPW), and camera (by reviewer YMMm), which are summarized in the following.
> >
> > - __4D Radar__: As discussed in the revised Appendix G (Experiments for the baseline network utilizing Doppler measurements), Doppler measurements representing the relative radial velocities of surrounding objects are useful for object detection. Therefore, in the revision, we provide additional 4D Radar baseline network (i.e., RTNH-Doppler) that utilize Doppler measurements in addition to the XYZ. Note that since we focus on the elevation information that is available in 4DRT but not in the conventional 3DRT, the main body of the paper discusses RTNH. However, discussions for RTNH-Doppler is added to Appendix G, and we open the RTNH-Doppler code to the public as well.
> >
> > - __Lidar__: As in Section 4, we compare the baseline network for 4D Radar to PointPillars that is a well-known baseline network for Lidar, which is because these two networks have similar structures such as (1) one-stage detector and (2) the same neck and head. However, as in the review comments, PointPillars (2019) is not new. To improve the 3D detection performance comparison of the proposed Lidar baseline networks, we select a recent network, PV-RCNN++ (2021), that has two-stage detector structure and we provide the training codes as well. (The training on the K-Radar dataset is ongoing, and we will post it on the project page as soon as the training is complete.)
> >
> > - __Camera__: We are going to upload the training codes and 3D detection performance for a recent camera-based 3D object detection baseline network, CaDNN (2021). Since CaDNN requires dense depth maps for training, we are generating depth maps, which can take about 2 months. As soon as the generation of the dense depth map completed, we are planning to train the network and provide the detection performance on the project page.

---

> > ### Author Response · Authors · 2022-08-20
> > **Response to (1) Calibration between camera and Lidar**
> >
> > __Q1__. However, the calibration part should be more detailed described, especially for camera and lidar extrinsics calibration.
> >
> > __A1__. Please refer to __[Response to (Common Issue #1) Calibration between camera and Lidar]__ in the __[Author response to the three common issues]__. Let us copy and paste our response below for your convenience:
> >
> > We agree that, in the original paper, the calibration between Lidar and 4D Radar is discussed in detail, but the calibration between Lidar and camera is not described enough. Therefore, in Appendix D.3 of the revision, we provide details of the full calibration procedure between Lidar and camera; from the lens distortion of the camera to the detailed calibration procedure and result between the Lidar and the camera.
> >
> > Summarizing the (added) calibration procedure in Appendix D.3, we first obtain coarse values of the intrinsic parameters (i.e., principal point, focal length, and lens distortion) and extrinsic parameters (i.e., rotation and translation between Lidar and camera) by using a checkerboard calibration provided in ROS [1] and 3D modeling of the sensor suite provided in iPhone 12 Pro [2], respectively. Then, we construct GUI (Graphical User Interfaces) to fine-tune these parameters, which enables us to reduce fine calibration errors (which [1] and [2] are difficult to guarantee) as seen in Figure 17 (added on page 8 of the revised Appendix). Finally, we show the accuracy of the calibration by illustrating the projected points onto the front image and generating depth maps as seen in Figure 18 and 19 (added on page 8 of the revised Appendix), respectively.
> >
> > Please note that the generated depth map is another contribution of our dataset; our dataset can be used as ground truth for camera-based depth estimation as well.
> >
> > [1] Stanford Artificial Intelligence Laboratory et al. Robotic operating system.
> >
> > [2] Gregor Luetzenburg, Aart Kroon, and Anders Bjørk. Evaluation of the apple iphone 12 pro lidar for an application in geosciences. Scientific Reports, 11, 2021.

---

### Author Response · Authors · 2022-08-20
**Author response to the three common issues**

First of all, we thank all of the reviewers for the valuable review comments.

In a summary, we have received review comments from 6 reviewers;

- 1 reviewer gave rating 8 (clear accept - vuZA)

- 2 reviewers gave rating 7 (accept - YMMm and zuKk).

- 2 reviewers gave rating 6 (marginally above acceptance threshold - fy92, and 6HK3)

- 1 reviewer gave rating 5 (marginally below acceptance threshold - 1XPW).

Despite the ratings scores in a wide range, we are so grateful to all of the reviewers for their valuable comments that greatly help us to improve the quality of our paper. Furthermore, we are particularly encouraged, because we receive very high rating scores by the three reviewers who have the highest confidence on their review.

Before we introduce our specific response to the review comments, let us list the three common issues in the review comments:

__(Issue #1) Calibration between camera and Lidar (by 3 reviewers: YMMm, zuKk, and fy92)__

__(Issue #2) Additional baseline networks (by 3 reviewers: YMMm, 1XPW, and vuZA)__

__(Issue #3) Missing datasets (by 2 reviewers: YMMm, and zuKk).__

Let us provide author responses to the above three common issues below:

---

> ### Author Response · Authors · 2022-08-20
> **Response to (Common Issue #3) Missing datasets**
>
> __[Q #3]__ In the review, two reviewers pointed out that
>
> “Missing related dataset: [1] Wang, Yizhou, et al. "Rethinking of Radar's Role: A Camera-Radar Dataset and Systematic Annotator via Coordinate Alignment." Proceedings of the IEEE/CVF Conference on Computer Vision and Pattern Recognition. 2021” by YMMm, and
>
> “The paper compared their dataset with other autonomous driving datasets and clearly state their advantages against others. The authors should consider including another radar object detection dataset RODNet [1] in their related works. [1] Wang, Yizhou, et al. "RODNet: A real-time radar object detection network cross-supervised by camera-radar fused object 3D localization." IEEE Journal of Selected Topics in Signal Processing 15.4 (2021): 954-967.” by zuKk.
>
> __[Answer to issue #3]__
>
> First of all, we thank the reviewers for notifying us of these datasets. In Table 1, and 2 of the revision, we add the missing related datasets, such as CRUW and VoD as follows.
>
> ***
>
> Table 1. Comparison of object detection datasets and benchmarks for autonomous driving. HR and LR refer to High Resolution Lidar with more than 64 channels and Low Resolution with less than 32 channels, respectively. Bbox., Tr.ID, and Odom. refer to bounding box annotation, tracking ID, and odometry, respectively. Bolded text indicates the best entry in each category.
> | Dataset | Num.data | RT | RPC | LPC | Camera | GPS | Bbox. | Tr.ID | Odom. |
> |:---:|:---:|:---:|:---:|:---:|:---:|:---:|:---:|:---:|:---:|
> | K-Radar (ours) | 35K | **4D** | **4D** | **HR.** | __360.__ | __RTK__ | __3D__ | __O__ | __O__ |
> | VoD | 8.7K | X | **4D** | **HR.** | Front | __RTK__ | __3D__ | __O__ | __O__ |
> | Astyx | 0.5K | X | **4D** | LR. | Front | X | __3D__ | X | X |
> | RADDet | 10K | 3D | 3D | X | Front | X | 2D | X | X |
> | Zendar | 4.8K | 3D | 3D | LR. | Front | GPS | 2D | __O__ | O |
> | RADIATE | 44K | 3D | 3D | LR. | Front | GPS | 2D | __O__ | O |
> | CARRADA | 12.6K | 3D | 3D | X | Front | X | 2D | __O__ | X |
> | CRUW | 396K | 3D | 3D | X | Front | X | Point | __O__ | X |
> | NuScenes | 40K | X | 3D | LR. | __360.__ | __RTK__ | __3D__ | __O__ | __O__ |
> | Waymo | 230K | X | X | **HR.** | z | X | __3D__ | __O__ | X |
> | KITTI | 15K | X | X | **HR.** | Front | __RTK__ | __3D__ | __O__ | __O__ |
> | BDD100k | **120M** | X | X | X | Front | __RTK__ | 2D | __O__ | __O__ |
>
> ***
>
> Table 2. Comparison of object detection datasets and benchmarks for autonomous driving. d/n refers to day and night. Bolded text indicates the best entry in each category.
> | Dataset | Weather conditions | Time |
> |:---:|:---:|:---:|
> | K-Radar (ours) | **overcast, fog, rain, sleet, snow** | **d/n** |
> | VoD | X | day |
> | Astyx | X | day |
> | RADDet | X | day |
> | Zendar | X | day |
> | RADIATE | overcast, fog, rain, snow | **d/n** |
> | CARRADA | X | day |
> | CRUW | X | day |
> | NuScenes | overcast, rain | **d/n** |
> | Waymo | overcast | **d/n** |
> | KITTI | X | day |
> | BDD100k | overcast, fog, rain, snow | **d/n** |
>
> ***
>
> In Section 2 of the revision (line 150 ~ 154), we also add discussion for CRUW as follows (copy and paste): “CRUW (Wang et al., 2021) provides a large number of 3DRTs, but annotations only provide 2D point locations of objects. VoD (Palffy et al., 2022) and Asytx (Meyer and Kuschk, 2019) provide 3D bounding box labels with 4DRPCs. However, the dense 4DRTs are not made available, and the number of samples in the datasets is relatively small (i.e., 8.7K and 0.5K frames).”
>
> Notice that, starting from the next page, we provide our response to each of the review comments per reviewer. However, for the review comments related to the above 3 common issues, we just provide a short notice that our response is in __[Author Response to the Three Common Issues]__ that is in the first two pages of this document. Thank you so much.

---

> ### Author Response · Authors · 2022-08-20
> **Response to (Common Issue #2) Additional baseline networks**
>
> __[Q #2]__ In the review, three reviewers pointed out that
>
> “Baseline and benchmarking are not sufficient. More radar-based baselines can be added. & Some other baselines are also recommended to be added like camera-based methods.” by YMMm,
>
> “The pointpillars is a pretty old baseline for lidar-based object detection. Why did the authors choose this baseline? Please quantitatively report the results of using newer baselines from 2021.” by 1XPW, and
>
> “It would be better if more methods are compared in the benchmarking” by vuZA.
>
> __[Answer to issue #2]__
>
> We agree with your comment. In the revision, we provide additional baseline networks for 4D Radar (indicated by reviewer YMMm), Lidar (by reviewer 1XPW), and camera (by reviewer YMMm), which are summarized in the following.
>
> - __4D Radar__: As discussed in the revised Appendix G (Experiments for the baseline network utilizing Doppler measurements), Doppler measurements representing the relative radial velocities of surrounding objects are useful for object detection. Therefore, in the revision, we provide additional 4D Radar baseline network (i.e., RTNH-Doppler) that utilize Doppler measurements in addition to the XYZ. Note that since we focus on the elevation information that is available in 4DRT but not in the conventional 3DRT, the main body of the paper discusses RTNH. However, discussions for RTNH-Doppler is added to Appendix G, and we open the RTNH-Doppler code to the public as well.
>
> - __Lidar__: As in Section 4, we compare the baseline network for 4D Radar to PointPillars that is a well-known baseline network for Lidar, which is because these two networks have similar structures such as (1) one-stage detector and (2) the same neck and head. However, as in the review comments, PointPillars (2019) is not new. To improve the 3D detection performance comparison of the proposed Lidar baseline networks, we select a recent network, PV-RCNN++ (2021), that has two-stage detector structure and we provide the training codes as well. (The training on the K-Radar dataset is ongoing, and we will post it on the project page as soon as the training is complete.)
>
> - __Camera__: We are going to upload the training codes and 3D detection performance for a recent camera-based 3D object detection baseline network, CaDNN (2021). Since CaDNN requires dense depth maps for training, we are generating depth maps, which can take about 2 months. As soon as the generation of the dense depth map completed, we are planning to train the network and provide the detection performance on the project page.

---

> ### Author Response · Authors · 2022-08-20
> **Response to (Common Issue #1) Calibration between camera and Lidar**
>
> __[Q #1]__ In the review, three reviewers pointed out that
>
> “However, the calibration part should be more detailed described, especially for camera and lidar extrinsics calibration.” by YMMm,
>
> “The calibration process for the sensor suite is not very clear. The paper has mentioned the calibration between radar and lidar but did not explain other extrinsic calibrations for cameras. The camera extrinsic calibration is important as they show the dash-camera image as a supporting image for annotation in the appendix.” by zuKk, and
>
> “The authors do not mention the calibration process of other sensors to Lidar except radar. The yellow bounding boxes shown in https://youtu.be/b_9TtOxaN1w at 1:47 seem not aligned with cars on the image due to the camera and lidar calibration issue.” by fy92.
>
> __[Answer to issue #1]__
>
> We agree that, in the original paper, the calibration between Lidar and 4D Radar is discussed in detail, but the calibration between Lidar and camera is not described enough. Therefore, in Appendix D.3 of the revision, we provide details of the full calibration procedure between Lidar and camera; from the lens distortion of the camera to the detailed calibration procedure and result between the Lidar and the camera.
>
> Summarizing the (added) calibration procedure in Appendix D.3, we first obtain coarse values of the intrinsic parameters (i.e., principal point, focal length, and lens distortion) and extrinsic parameters (i.e., rotation and translation between Lidar and camera) by using a checkerboard calibration provided in ROS [1] and 3D modeling of the sensor suite provided in iPhone 12 Pro [2], respectively. Then, we construct GUI (Graphical User Interfaces) to fine-tune these parameters, which enables us to reduce fine calibration errors (which [1] and [2] are difficult to guarantee) as seen in Figure 17 (added on page 8 of the revised Appendix). Finally, we show the accuracy of the calibration by illustrating the projected points onto the front image and generating depth maps as seen in Figure 18 and 19 (added on page 8 of the revised Appendix), respectively.
>
> Please note that the generated depth map is another contribution of our dataset; our dataset can be used as ground truth for camera-based depth estimation as well.
>
> [1] Stanford Artificial Intelligence Laboratory et al. Robotic operating system.
>
> [2] Gregor Luetzenburg, Aart Kroon, and Anders Bjørk. Evaluation of the apple iphone 12 pro lidar for an application in geosciences. Scientific Reports, 11, 2021.

---

### Meta-Review · Area_Chair_yn8j · 2022-09-02

**Recommendation:** Accept
**Confidence:** 4

**Metareview:**

5 out of 6 reviewers are positive about accepting this work. The meta reviewer agrees with the reviewers that the K-Radar dataset is a clear contribution to the community and thus recommends acceptance. The meta reviewer believes that the authors have answered the negative reviewer's question in a fair enough way. The authors are suggested the authors polish their paper writing and exposition following reviewers' comments.

---

### Decision · Program_Chairs · 2022-09-16

Accept